

# Non-boost invariant fluid dynamics

**Jan de Boer[1], Jelle Hartong[2], Emil Have[2], Niels A. Obers[3,4] and Watse Sybesma[5]**

**1** Institute for Theoretical Physics and Delta Institute for Theoretical Physics,
University of Amsterdam, Science Park 904, 1098 XH Amsterdam, The Netherlands
**2** School of Mathematics and Maxwell Institute for Mathematical Sciences,
University of Edinburgh, Peter Guthrie Tait Road, Edinburgh EH9 3FD, UK
**3** Nordita, KTH Royal Institute of Technology and Stockholm University,
Roslagstullsbacken 23, SE-106 91 Stockholm, Sweden
**4** The Niels Bohr Institute, Copenhagen University,
Blegdamsvej 17, DK-2100 Copenhagen Ø, Denmark
**5** University of Iceland, Science Institute, Dunhaga 3, IS-107, Reykjavík, Iceland

## Abstract

We consider uncharged fluids without any boost symmetry on an arbitrary curved background and classify all allowed transport coefficients up to first order in derivatives. We assume rotational symmetry and we use the entropy current formalism. The curved background geometry in the absence of boost symmetry is called absolute or Aristotelian spacetime. We present a closed-form expression for the energy-momentum tensor in Landau frame which splits into three parts: a dissipative (10), a hydrostatic non-dissipative (2) and a non-hydrostatic non-dissipative part (4), where in parenthesis we have indicated the number of allowed transport coefficients. The non-hydrostatic non-dissipative transport coefficients can be thought of as the generalization of coefficients that would vanish if we were to restrict to linearized perturbations and impose the Onsager relations. For the two hydrostatic and the four non-hydrostatic non-dissipative transport coefficients we present a Lagrangian description. Finally when we impose scale invariance, thus restricting to Lifshitz fluids, we find 7 dissipative, 1 hydrostatic and 2 non-hydrostatic non-dissipative transport coefficients.



# 1   Introduction

Hydrodynamics arises as the universal description of interacting systems near local thermal equilibrium in the long wavelength limit, which makes hydrodynamics indispensable as an effective theory for a broad class of physical phenomena. Once a hydrodynamic description is established for a system, its evolution is governed by the hydrodynamic equations of motion, which express the conservation of currents such as the energy-momentum tensor. The relevant currents are parametrized in terms of fluid variables such as temperature, fluid velocity and chemical potential via the constitutive relations – see e.g. [1, 2].

In the standard treatment of hydrodynamical frameworks some type of boost symmetry is assumed, namely Galilean boost symmetry for non-relativistic hydrodynamics and Lorentz boost symmetry for its relativistic counterpart. Consequently, these symmetries are present in the well-known Navier-Stokes equations [1] and relativistic hydrodynamics [2] or magneto-hydrodynamics [3–6] respectively. While these boost symmetries are conventionally assumed, there is no a priori reason to require any type of boost symmetry in the formulation of hydrodynamics.[1] From a theoretical point of view they are not necessary as the hydrodynamic equations generally follow from conservation of energy/momentum and other charges, which in turn are connected to time/space translations and possible extra global symmetries. Boost symmetries, on the other hand, provide relations between components of the various currents, and are as such not an essential ingredient, though they are reflected as extra symmetries of the resulting hydrodynamic equations. In fact, as we will return to in more detail shortly, there exist many physical systems that do not exhibit boost symmetry. In particular, as soon as there is a preferred reference frame, i.e. a medium with respect to which the fluid moves,

---

[1]Assuming spatial rotational symmetry is not necessary either, though often taken as an extra symmetry as is also the case in the present work. Hydrodynamics for anisotropic systems has been studied in several places, e.g. in [7, 8].

this symmetry will be broken. Moreover, breaking of boost symmetry also occurs in critical systems with scaling symmetry characterized by a generic dynamical exponent $z$, i.e. systems with Lifshitz symmetry. Importantly, the no-go theorem of [9] says that, when $z \neq 1, 2$, such systems cannot exhibit boost symmetry if they allow for a fluid description. A similar no-go theorem was found in Ref. [10] from a field theoretic point of view.

A systematic treatment of perfect fluids with translation and rotation symmetries, applicable in the absence of any type of boost symmetry was first given in [9]. In a subsequent work [11] the first-order hydrodynamics and transport for these perfect fluids was studied when linearizing around a zero velocity background.

The primary aim of this paper is to complete this first-order analysis, as announced in [11], by extending it to the full non-linear level. In particular, we will analyze such fluids up to first order in derivatives for arbitrary fluid velocity backgrounds on curved absolute spacetime and find all dissipative and non-dissipative transport coefficients. Moreover, among the latter we will identify those that are hydrostatic and those that are not. We remark that, following the works [9, 11], first-order corrections to non-boost invariant hydrodynamics on flat space were also pursued in Ref. [12] (which also includes a $U(1)$ current).[2] Furthermore, hydrodynamics of systems without boosts has been addressed previously (see e.g. [13, 14] and [15–17]) but our starting point, the perfect fluid thermodynamics introduced in [9], differs from these works.

Our hydrodynamic description starts from the perfect fluid energy-momentum tensor for non-boost invariant systems obtained in Ref. [9]. This includes a new thermodynamic variable, the kinetic mass density $\rho$, which is the thermodynamic dual of the magnitude of the fluid velocity squared, $v^2$. Furthermore, as an immediate consequence of the absence of boost symmetry, all extensive thermodynamic quantities now also depend on the extra intrinsic thermodynamic quantity $v$. The analysis of [9] shows that this more general class of perfect fluids leads to corrections to the Euler equations, which might be observable in hydrodynamic fluid experiments. One also finds new expressions for the speed of sound in perfect fluids, reducing to known results when boost symmetry is present. A concrete realization of this framework can be obtained by considering an ideal gas of Lifshitz particles, enabling for example to obtain expressions for the speed of sound for corresponding classical and quantum Lifshitz gases [9]. Furthermore, the linearized first-order analysis in [11] has provided novel expressions for the linearized Navier–Stokes equation including new dissipative and non-dissipative first order transport coefficients.

As is well known, in order to account for dissipative effects, the conserved currents entering the effective description are expanded to a given order in derivatives of the hydrodynamic fields under the assumption that these derivative corrections are small compared to some intrinsic length scale of the microscopic system (e.g. the mean free path). In the currents – and therefore also in the resulting hydrodynamic equations of motion – each independent derivative correction term is multiplied by a transport coefficient, such as viscosity and conductivity. The values of these coefficients are constrained by the requirement that the divergence of the entropy current is non-negative and, additionally, by the Onsager relations (or, rather, their appropriate generalization: absence of anti-symmetric transport) in systems with time reversal symmetry. For systems with a microscopic description, the specific form of certain transport coefficients can be determined via Kubo formulae. If a system admits a gravitational dual, further relations abound: notably, it has been shown via the AdS/CFT correspondence that shear viscosity divided by entropy density is equal to $\frac{1}{4\pi}$ for a strongly coupled plasma in $\mathcal{N} = 4$

---

[2]At the level of constitutive relations the present work agrees with [12]. However comparing the implications of the non-negativity of entropy production and the properties of the non-dissipative transport coefficients requires one to extract the necessary details from the accompanying Mathematica notebooks of Ref. [12], making it challenging to perform an extensive mapping of our final results.

supersymmetric Yang–Mills theory [18].

To obtain the first-order hydrodynamics for non-boost invariant systems[3] following the paradigm reviewed above, the analysis of this paper makes crucially use of the geometry that is connected to non-boost invariant fluids along with the power of hydrostatic partition functions and the entropy current formalism. The geometry on which non-boost invariant fluids live is the geometry that realizes (locally) only translational (space and time) and rotational symmetries.[4] These symmetries are sometimes called *Aristotelian* and the corresponding geometry is that of curved absolute spacetime, which thus can also be referred to as Aristotelian geometry.

An interesting feature of non-boost invariant fluids is the appearance of non-dissipative transport coefficients at first order, alongside dissipative transport coefficients [11, 12]. By applying the entropy current constraint to the full non-linear constitutive relations we show in this paper that there are 10 dissipative transport coefficients and 6 non-dissipative ones. We also show that the number of transport coefficients is unaffected by the introduction of background curvature to first order in derivatives. Following [25, 26], one can further separate the non-dissipative ones into two types, hydrostatic and non-hydrostatic, which in the present case turns out to be 2 and 4 transport coefficients, respectively. For the case of Lifshitz fluids, these numbers become 7, 1 and 2, respectively. We will show that both the hydrostatic and non-hydrostatic transport coefficients can be obtained using Lagrangian methods. The hydrostatic transport coefficients feature in the non-canonical part of the entropy current and coincide with contributions that can be computed using an action principle obtained by allowing for time dependence in the hydrostatic partition function, see e.g. [27, 28] and the earlier works [29, 30]. Furthermore, we find that when restricting to linearized perturbations all the non-hydrostatic transport coefficients vanish due to the Onsager relations.

We now return to a brief discussion of the physical relevance of non-boost invariant hydrodynamics, before presenting an outline of the paper.

**Relevance of non-boost invariant hydrodynamics**

As remarked above, for many systems in nature one does not have the luxury of assuming boost symmetry. In condensed matter, for example, one can study critical points where boost symmetries are absent [31]. The Lifshitz critical point [32] is an example of this and related recent papers include quantum critical transport in strange metals, see e.g. [33], electrons in graphene [34] and viscous electron fluids [35]. With this application in mind, it is shown in the original Refs. [9, 11] how the framework of non-boost invariant hydrodynamics can be adapted to (non-relativistic) scale invariant fluids with critical exponent $z$. This includes particular expressions for the speed of sound in generic $z$ Lifshitz fluids as well as specific results for the first-order transport coefficients in the linearized case. In particular, it was shown that the sound attenuation constant depends on both shear viscosity and thermal conductivity. The framework was also recently used in [36] to study out-of-equilibrium energy transport in a quantum critical fluid with Lifshitz scaling symmetry following a local quench between two semi-infinite fluid reservoirs. It is also interesting to note that Lifshitz hydrodynamics is relevant in connection with non-AdS holographic realizations of systems with Lifshitz thermodynamics [23, 37–40], see also [41–45].

More generally non-boost invariant hydrodynamics is of relevance to any system with a reference frame, such as æther theories or in various active matter systems exhibiting e.g.

---

[3]In the way we set up our calculations, we consider fluids which could have relativistic or Galilean boosts. The Carrollian boost invariant fluid as realized in e.g. [9, 19] will not be considered in the present work. This specific situation will be treated in [20].

[4]Similarly, non-relativistic (Galilean) fluids live on the geometry that locally realizes these symmetries and in addition Galilean boosts, i.e. Newton-Cartan geometry. This was used in e.g. [19, 21–24].

flocking behavior, see e.g. [46] and [47] for a recent example. General active matter systems typically do not have conserved energy or momentum as the divergence of the energy and momentum currents is equal to 'driving' terms. Non-boost invariant hydrodynamics is only an approximate description for configurations that are close to equilibrium configurations at 'cruising speed' where the driving terms vanish.

Assuming that the fundamental laws of physics are Lorentz invariant, the necessity of some type of reference frame to obtain a system with broken boosts is obvious. Dispersion relations which are non-analytic and incompatible with boost invariance, such as those of capillary waves and domain-wall fluctuations in superfluid interfaces (ripplons), see for example [48], do indeed describe the propagation of particular fluctuations with respect to a medium. But in order to have a hydrodynamic description of excitations with respect to a medium, we also need that energy and momentum of these excitations are approximately conserved. This is a non-trivial requirement, and requires a relatively weak coupling of the excitations to the medium. In addition, in order to be in the hydrodynamic regime, the interaction times and length scales of the excitations with themselves must be much smaller than those of the excitations with the medium. For example, for electrons in a crystal, the electron-electron scattering rate must be much higher than the rate for scattering of impurities and phonons in order to possibly have hydrodynamic flow [34]. Obviously, it is an extremely interesting question to find physical systems where all these conditions apply, which we will not address in this paper.

It is of separate interest to understand systems with broken boost symmetry from an effective field theory point of view. Here, the boost breaking arises because we integrate out the degrees of freedom of the medium in a state which breaks the boost symmetry, for example because the medium has a fixed density or particle number. In [49] a classification of condensed matter systems that break boost symmetry but preserve rotation and translation invariance was presented. The simplest possibility presented there was a so-called type I framid, which is a system where the unbroken spacetime translation and rotation generators are unmodified in the presence of a boost-breaking state. Curiously, this possibility does not seem to be realized in nature, which was also seen in a recent analysis of the structure of Goldstone bosons associated to boost breaking [50]. In such a putative type I framid the expectation value of the energy-momentum tensor must be proportional to the metric with the sum of the energy density and the pressure equal to zero. This is similar to the effective energy-momentum tensor one can associate to a cosmological constant, and also the form of the energy-momentum tensor in the presence of additional 'Carroll' boost invariance [9]. It is unclear whether any systems in nature properly realize Carroll symmetry and this observation may be in fact equivalent to the non-existence of type I framids. For a more detailed discussion of systems with (approximate) Carroll symmetry, we refer the reader to [20].

A simple example which gives rise to a system without boost invariance and which does appear in nature is a superfluid with a spontaneously broken $U(1)$ symmetry of the type considered in [51]. These systems contain a coupling $\int d^{d+1}x A_\mu J^\mu$ where $J^\mu$ is the current associated to the global $U(1)$ symmetry and $A_\mu$ acquires an expectation value $A_\mu = \lambda \delta^0_\mu$. This is reminiscent of a chemical potential and we assume that at finite $\lambda$ the ground state breaks the global $U(1)$ symmetry. If $\lambda$ is constant the system remains invariant under translations and rotations. Our hydrodynamical description will still be a good approximation as long as $|\partial_\mu \lambda| \ll |\lambda|$ so that energy and momentum are approximately conserved. To find the energy-momentum tensor of the theory, it is convenient to use vielbeins to convert curved into flat indices, and to assume that $A_a = \lambda \delta^0_a$. This is now a scalar and not a tensor from the spacetime point of view, and one can obtain a conserved stress-tensor by varying with respect to the vielbeins as described in e.g. [52]. The result of this computation is a new stress tensor of the form $T^{new}_{\mu\nu} \sim T^{old}_{\mu\nu} + J_\mu A_\nu$ which is clearly not symmetric and therefore incompatible with boost invariance. Moreover it shows that the new generator of time translations is a linear

combination of the old generator plus the $U(1)$ current. As described in detail in [49], other breaking patterns are also possible.

**Outline**

The paper is organized in the following way. In Section 2, we recap and establish facts about perfect boost-agnostic[5] fluids and Aristotelian geometry. Next, in Section 3, we formulate the entropy current on curved spacetime and define three sectors of transport: hydrostatic non-dissipative, non-hydrostatic non-dissipative and dissipative transport. Subsequently, in Section 4, we present a Lagrangian description of both hydrostatic and non-hydrostatic transport and furthermore we connect the Lagrangian formulation to the non-canonical contributions to the entropy current. Finally, in Section 5 we obtain and combine all first order transport coefficients using constitutive relations. We end with a discussion and outlook in Section 6. Two appendices are included. Appendix A presents the features of hydrodynamic frame transformations from a generic frame to the Landau frame. Appendix B presents the details of the derivation of the non-canonical entropy current by using constitutive relations (as opposed to an action formulation as dones in Section 4).

## 2 Non-boost invariant perfect fluids and geometry

### 2.1 Perfect fluids on flat spacetime

Consider a charged perfect fluid in $(d+1)$-dimensions, which has spatial rotational invariance and translational invariance in both time and space, as was studied in [9,11]. One can choose the fluid variables to be chemical potential $\mu$, temperature $T$ and fluid velocity $v^i$. These variables are allowed to depend on space and time. It is assumed, however, that locally there exists a thermodynamic equilibrium. We furthermore assume pressure $P$ to be a function of $\mu$, $T$ and $v^2$. In other words, we assume the equation of state to be of the form $P \equiv P(T, \mu, v^2)$. Through the Gibbs-Duhem relation,

$$dP = s\,dT + n\,d\mu + \frac{1}{2}\rho\,dv^2\,, \tag{2.1}$$

we can express entropy density $s$, charge density $n$ and kinetic mass density $\rho$ in terms of the fluid variables. Finally, we have the Euler relation

$$\mathcal{E} = -P + sT + \mu n + \rho v^2\,, \tag{2.2}$$

which expresses the total energy density $\mathcal{E}$ in terms of the fluid variables. In the presence of a boost symmetry, the corresponding Ward identity implies that the term containing kinetic mass density in (2.1) and (2.2) can be absorbed into the other fluid variables [9,11]. This leads to velocity independent thermodynamic relations – a hallmark of boost invariant systems.

The conserved currents, energy-momentum tensor $T^\mu{}_\nu$ and charge current $J^\mu$ can all be expressed as functions of the fluid variables in a derivative expansion, the form of which is dictated by the symmetry of the problem. The divergence of these conserved currents gives rise to the dynamics of the fluid. The most general homogeneous and isotropic perfect fluid, i.e. at zeroth order in derivatives, is characterized by [9,11]

$$T^0{}_0 = -\mathcal{E}\,, \quad T^i{}_0 = -(\mathcal{E}+P)v^i\,, \quad T^0{}_j = \rho v_j\,, \quad T^i{}_j = P\delta^i{}_j + \rho v^i v_j\,, \tag{2.3}$$

---

[5]We sometimes use the terminology 'boost-agnostic' (instead of non-boost invariant) to highlight the fact that the analysis holds in principle for any fluid regardless of whether it has boost symmetries or not.

$$J^0 = n \,, \quad J^i = n v^i \,. \tag{2.4}$$

Here we presented the fluid written in the LAB frame, in which the observer is at rest. For fluids without boost symmetries, it is useful to define the internal energy

$$\tilde{\mathcal{E}} = \mathcal{E} - \rho v^2 \,, \tag{2.5}$$

in terms of which the Euler and Gibbs-Duhem relations read

$$\tilde{\mathcal{E}} + P = T s + \mu n \,, \qquad d\tilde{\mathcal{E}} + \frac{1}{2}\rho d v^2 = T ds + \mu dn \,, \qquad dP = s dT + n d\mu + \frac{1}{2}\rho d v^2 \,. \tag{2.6}$$

In the subsequent analysis, we will drop the charge current.

## 2.2 Curved geometry for non-boost invariant fluids

Fluids without boosts live on the geometry of absolute spacetime, which, due to Penrose, is sometimes referred to as *Aristotelian* geometry [53]. This geometry locally realizes Aristotelian symmetries consisting of translations in time and space along with rotations.

We define an Aristotelian geometry in terms of a 1-form $\tau_\mu$ (clock-form), along with a symmetric covariant tensor $h_{\mu\nu}$ (spatial metric), where, notably, neither field has been assigned any local tangent space transformations. The signature of $h_{\mu\nu}$ is $(0, 1, \dots, 1)$. We emphasize that this is different from (torsional) Newton–Cartan geometry (see e.g. [54–56]), in which $h_{\mu\nu}$ is endowed with transformation properties corresponding to local Galilean boosts. Likewise, in Carrollian geometry (see for example [57]), it is $\tau_\mu$ – but not $h_{\mu\nu}$ – that transforms under local Carrollian boosts. In the more familiar case of Lorentzian geometry, both $\tau_\mu$ and $h_{\mu\nu}$ transform under local Lorentz transformations in such a way that $\gamma_{\mu\nu} = -\tau_\mu \tau_\nu + h_{\mu\nu}$ remains invariant.

In this way, Galilean[6], Carrollian, and Lorentzian geometries all arise as special cases of Aristotelian geometry via the imposition of specific local tangent space transformations on the geometric data $\tau_\mu$ and $h_{\mu\nu}$.

Because of the signature of $h_{\mu\nu}$, we can decompose it into vielbeins in the following manner

$$h_{\mu\nu} = \delta_{ab} e_\mu^a e_\nu^b \,, \tag{2.7}$$

where $a = 1, \dots, d$ and $\mu$ takes $d+1$ values. The spatial vielbeins $e_\mu^a$ transform under local $SO(d)$ transformations. Note that the square matrix $(\tau_\mu, e_\mu^a)$ is invertible with inverse denoted by $(v^\mu, e_a^\mu)$ – these objects satisfy the following orthonormality relations

$$v^\mu \tau_\mu = -1 \,, \qquad v^\mu e_\mu^a = 0 \,, \qquad e_a^\mu \tau_\mu = 0 \,, \qquad e_a^\mu e_\mu^b = \delta_a^b \,. \tag{2.8}$$

Furthermore, these fields satisfy the completeness relation

$$-v^\mu \tau_\nu + e_a^\mu e_\nu^a = \delta_\nu^\mu \,. \tag{2.9}$$

The determinant of $(\tau_\mu, e_\mu^a)$ will be denoted by $e$, i.e.

$$e = \det(\tau_\mu, e_\mu^a) \,. \tag{2.10}$$

For completeness, we remark that it is in general possible to choose an affine connection $\Gamma_{\mu\nu}^\lambda$ that obeys

$$\nabla_\mu \tau_\nu = 0 \,, \qquad \nabla_\mu h_{\nu\rho} = 0 \,. \tag{2.11}$$

---

[6]We remark that in order to obtain torsional Newton–Cartan geometry from Aristotelian geometry, we need an extra gauge field. See e.g. [56, 58] and references therein.

Let us make the following ansatz for $\Gamma^{\lambda}_{\mu\nu}$:

$$\Gamma^{\lambda}_{\mu\nu} = -v^{\lambda}\partial_{\mu}\tau_{\nu} + \frac{1}{2}h^{\lambda\kappa}\left(\partial_{\mu}h_{\nu\kappa} + \partial_{\nu}h_{\mu\kappa} - \partial_{\kappa}h_{\mu\nu}\right) - h^{\lambda\kappa}\tau_{\nu}K_{\mu\kappa} + C^{\lambda}_{\mu\nu}, \tag{2.12}$$

where we introduced the extrinsic curvature

$$K_{\mu\nu} = -\frac{1}{2}\mathcal{L}_{\nu}h_{\mu\nu} . \tag{2.13}$$

The extrinsic curvature is purely spatial,

$$v^{\mu}K_{\mu\nu} = 0 , \tag{2.14}$$

and its trace satisfies

$$K := h^{\mu\nu}K_{\mu\nu} = -e^{-1}\partial_{\mu}(ev^{\mu}) , \quad h^{\mu\nu} = \delta^{ab}e^{\mu}_{a}e^{\nu}_{b} . \tag{2.15}$$

The equations (2.11) are obeyed if we take

$$C^{\lambda}_{\mu\nu}\tau_{\lambda} = 0, \qquad C^{\lambda}_{\mu\nu}h_{\lambda\rho} + C^{\lambda}_{\mu\rho}h_{\nu\lambda} = 0. \tag{2.16}$$

We can for example choose a connection with $C^{\lambda}_{\mu\nu} = 0$. This connection has non-zero torsion given by

$$\Gamma^{\lambda}_{[\mu\nu]} = -\frac{1}{2}v^{\lambda}\tau_{\mu\nu} + \frac{1}{4}h^{\lambda\kappa}\tau_{\nu}\mathcal{L}_{\nu}h_{\mu\kappa} - \frac{1}{4}h^{\lambda\kappa}\tau_{\mu}\mathcal{L}_{\nu}h_{\nu\kappa}, \tag{2.17}$$

where we defined the torsion 2-form

$$\tau_{\mu\nu} = \partial_{\mu}\tau_{\nu} - \partial_{\nu}\tau_{\mu} . \tag{2.18}$$

We have included the discussion of the connection for completeness. However, we will never use any particular connection in this paper. Up to first order in derivatives we can make do with Lie derivatives, exterior derivatives and divergences. At second order in the derivative expansion, however, a choice must be made in order to write down curvatures.

Finally, we remark that flat Aristotelian spacetime in Cartesian coordinates corresponds to

$$\tau_{\mu} = \delta^{0}_{\mu}, \qquad h_{\mu\nu} = \delta^{i}_{\mu}\delta^{i}_{\nu}, \qquad v^{\mu} = -\delta^{\mu}_{0}, \qquad h^{\mu\nu} = \delta^{\mu}_{i}\delta^{\nu}_{i} , \tag{2.19}$$

where we split the spacetime index $\mu = (0, i)$ into temporal and spatial directions.

## 2.3 Perfect fluids on a curved background

Consider a perfect fluid living on an Aristotelian geometry described by $\{\tau_{\mu}, h_{\mu\nu}\}$ with fluid velocity $u^{\mu}$ satisfying

$$u^{\mu}\tau_{\mu} = 1 . \tag{2.20}$$

On flat space, $u^{\mu} = (1, v^{i})$, and the curved space analogue of $v^{2}$ is $u^{2} = h_{\mu\nu}u^{\mu}u^{\nu}$. When combined with the completeness relation (2.9) this implies the following decomposition of the fluid velocity $u^{\mu}$,

$$u^{\mu} = -v^{\mu} + h^{\mu\rho}h_{\rho\nu}u^{\nu} , \tag{2.21}$$

in terms of timelike and spacelike components, respectively. The perfect fluid energy-momentum tensor (2.3) generalized to a curved background reads [9] (see also Section 4.2)

$$T^{\mu}{}_{\nu} = -\left(\tilde{\mathcal{E}} + P + \rho u^{2}\right)u^{\mu}\tau_{\nu} + \rho u^{\mu}u^{\rho}h_{\rho\nu} + P\delta^{\mu}_{\nu} . \tag{2.22}$$

Using the property (2.20), this can also be written as

$$T^\mu_{\ \nu} = u^\mu u^\rho \left[ -\left(\tilde{\mathcal{E}} + P + \rho u^2\right) \tau_\rho \tau_\nu + \rho h_{\rho\nu} \right] + P \delta^\mu_\nu \,. \tag{2.23}$$

It is furthermore useful to express the energy-momentum tensor $T^\mu_{\ \nu}$ as

$$T^\mu_{\ \nu} = -T^\mu \tau_\nu + T^{\mu\rho} h_{\rho\nu} \,, \tag{2.24}$$

where

$$T^\mu = \mathcal{E} u^\mu + P h^{\mu\rho} h_{\rho\nu} u^\nu \,, \tag{2.25}$$

$$T^{\mu\nu} = P h^{\mu\nu} + \rho u^\mu u^\nu \,, \tag{2.26}$$

are the energy current and momentum-stress tensor, respectively.

If we are dealing with a theory on a generic curved Aristotelian background for which there is an action principle, then diffeomorphism invariance implies the following conservation equation for the energy-momentum tensor[7]

$$e^{-1} \partial_\mu \left(e T^\mu_{\ \rho}\right) + T^\mu \partial_\rho \tau_\mu - \frac{1}{2} T^{\mu\nu} \partial_\rho h_{\mu\nu} = 0 \,. \tag{2.27}$$

This will be shown in Section 4.2. When we are dealing with an on-shell theory (as we are in the case of dissipative fluids) then we simply impose (2.27) as the correct conservation equation. This is the analogue of declaring $\nabla_\mu T^{\mu\nu} = 0$ to be the conservation equation in the relativistic case. Notice that in flat space (in Cartesian coordinates), Eq. (2.27) reproduces the usual divergence of the energy-momentum tensor, as it is supposed to. The first term expresses the usual divergence of the energy-momentum tensor, while the remaining terms are currents contracted with their sources, on which a derivative acts. It thus takes the standard form of a divergence of a current being equal to the sum of the responses times the derivative of the sources. For completeness, we remark that the equation of motion (2.27) admits the covariantization

$$\nabla_\mu T^\mu_{\ \rho} + T^\mu \nabla_\rho \tau_\mu - \frac{1}{2} T^{\mu\sigma} \nabla_\rho h_{\mu\sigma} - \left(\Gamma^\mu_{\mu\sigma} - e^{-1} \partial_\sigma e\right) T^\sigma_{\ \rho} + 2\Gamma^\lambda_{[\mu\rho]} T^\mu_{\ \lambda} = 0 \,, \tag{2.28}$$

for *any* choice of connection $\Gamma^\rho_{\mu\nu}$.

It is useful to recast the equation of motion (2.27) for the perfect fluid (2.25)–(2.26), using the thermodynamic relations (2.6), in the following form

$$0 = \mathcal{L}_\beta s + \frac{s}{2} h^{\mu\nu} \left(\mathcal{L}_\beta h_{\mu\nu} - h_{\mu\sigma} u^\sigma \mathcal{L}_\beta \tau_\nu - h_{\nu\sigma} u^\sigma \mathcal{L}_\beta \tau_\mu\right) \,, \tag{2.29}$$

$$0 = \Pi^\rho_{\ \nu} h_{\rho\lambda} \Big[ -sT h^{\lambda\mu} \mathcal{L}_\beta \tau_\mu + u^\lambda \mathcal{L}_\beta \rho$$
$$+ \frac{\rho}{2} \left(u^\mu h^{\kappa\lambda} + u^\kappa h^{\mu\lambda} + u^\lambda h^{\mu\kappa}\right) \left(\mathcal{L}_\beta h_{\kappa\mu} - h_{\kappa\sigma} u^\sigma \mathcal{L}_\beta \tau_\mu - h_{\mu\sigma} u^\sigma \mathcal{L}_\beta \tau_\kappa\right) \Big] \,, \tag{2.30}$$

where we introduced the spatial projector

$$\Pi^\rho_{\ \mu} = \delta^\rho_\mu - u^\rho \tau_\mu \,, \tag{2.31}$$

---

[7]Contracting (2.27) with a vector $k^\rho$, we get

$$0 = e^{-1} \partial_\mu \left(e k^\rho T^\mu_{\ \rho}\right) + T^\mu \mathcal{L}_k \tau_\mu - \frac{1}{2} T^{\mu\nu} \mathcal{L}_k h_{\mu\nu} \,,$$

so if $k^\rho$ is Killing (cf. Eqs. (4.2) and (4.3)), we find that $0 = e^{-1} \partial_\mu \left(e k^\rho T^\mu_{\ \rho}\right)$, that is to say, $k^\rho T^\mu_{\ \rho}$ is a conserved current.

which satisfies

$$\Pi^{\rho}{}_{\mu}u^{\mu} = 0 = \Pi^{\rho}{}_{\mu}\tau_{\rho} \,, \quad \Pi^{\mu}{}_{\alpha}\Pi^{\alpha}{}_{\nu} = \Pi^{\mu}{}_{\nu} \,. \tag{2.32}$$

Furthermore, the Lie-derivative $\mathcal{L}_{\beta}$ in (2.29) and (2.30) is defined with respect to the vector

$$\beta^{\mu} = u^{\mu}/T \,. \tag{2.33}$$

In the remainder of this section we establish some identities that will be crucial later on. Equation (2.29) expresses entropy conservation and (2.30) represents conservation of momentum. Using that $s$ and $\rho$ can both be thought of as functions of $T$ and $u^2$ along with the identities

$$u^{\mu}\mathcal{L}_{\beta}\tau_{\mu} = -\frac{1}{T^2}u^{\mu}\partial_{\mu}T \,, \tag{2.34}$$

$$u^{\mu}u^{\nu}\mathcal{L}_{\beta}h_{\mu\nu} = \frac{1}{T}u^{\mu}\partial_{\mu}u^2 - 2\frac{u^2}{T^2}u^{\mu}\partial_{\mu}T \,, \tag{2.35}$$

where $u^2 = h_{\mu\nu}u^{\mu}u^{\nu}$, we can view the perfect fluid equation as providing $\mathcal{L}_{\beta}\tau_{\rho}$ in terms of $\mathcal{L}_{\beta}h_{\mu\nu} - h_{\mu\sigma}u^{\sigma}\mathcal{L}_{\beta}\tau_{\nu} - h_{\nu\sigma}u^{\sigma}\mathcal{L}_{\beta}\tau_{\mu}$, i.e.

$$\mathcal{L}_{\beta}\tau_{\rho} = \frac{1}{2}X_{\rho}{}^{\mu\nu}(\mathcal{L}_{\beta}h_{\mu\nu} - h_{\mu\sigma}u^{\sigma}\mathcal{L}_{\beta}\tau_{\nu} - h_{\nu\sigma}u^{\sigma}\mathcal{L}_{\beta}\tau_{\mu}) \,, \tag{2.36}$$

where $X_{\rho}{}^{\mu\nu}$ is given by

$$\begin{aligned}
X_{\rho}{}^{\mu\nu} &= \frac{1}{sT}\Pi^{\sigma}{}_{\rho}\left\{ 2\left(\frac{\partial P}{\partial u^2}\right)_s h_{\sigma\lambda}u^{\lambda}h^{\mu\nu} + 2\left(\frac{\partial \rho}{\partial u^2}\right)_s h_{\sigma\lambda}u^{\lambda}u^{\mu}u^{\nu} + \rho\left(u^{\mu}\delta^{\nu}_{\sigma} + u^{\nu}\delta^{\mu}_{\sigma}\right) \right\} \\
&\quad + \tau_{\rho}\frac{1}{T}\left[ u^{\mu}u^{\nu}\left(\frac{\partial \rho}{\partial s}\right)_{u^2} + h^{\mu\nu}\left(\frac{\partial P}{\partial s}\right)_{u^2} \right] \,.
\end{aligned} \tag{2.37}$$

The expression in parentheses on the RHS of (2.36) has the nice property that

$$\mathcal{L}_{\beta}h_{\mu\nu} - h_{\mu\sigma}u^{\sigma}\mathcal{L}_{\beta}\tau_{\nu} - h_{\nu\sigma}u^{\sigma}\mathcal{L}_{\beta}\tau_{\mu} = \frac{1}{T}\left( \mathcal{L}_u h_{\mu\nu} - h_{\mu\sigma}u^{\sigma}\mathcal{L}_u\tau_{\nu} - h_{\nu\sigma}u^{\sigma}\mathcal{L}_u\tau_{\mu} \right) \,, \tag{2.38}$$

so that these are Lie derivatives along velocity with, notably, an absence of $T$ derivatives. Finally, we remark that on flat space (cf. Eq. (2.19)), this special combination reduces to

$$\mathcal{L}_{\beta}h_{\mu\nu} - h_{\mu\sigma}u^{\sigma}\mathcal{L}_{\beta}\tau_{\nu} - h_{\nu\sigma}u^{\sigma}\mathcal{L}_{\beta}\tau_{\mu} \xrightarrow{\text{flat}} \frac{1}{T}\left( h_{\mu\sigma}\partial_{\nu}u^{\sigma} + h_{\nu\sigma}\partial_{\mu}u^{\sigma} \right) \,, \tag{2.39}$$

which will be useful in Section 5, where – after setting up the general problem of first-order corrections to non-boost invariant fluids – we specialize to flat space.

## 3 Entropy current

Going beyond perfect fluids means moving away from local thermodynamic equilibrium and requires derivative corrections to be added to the energy-momentum tensor. We will work up to first order in derivatives. The goal will be to identify all allowed tensorial structures whose coefficients are known as transport coefficients. By 'allowed' we mean 'allowed by symmetry' and furthermore 'allowed by entropy considerations'. We will impose that the fluid locally obeys the second law of thermodynamics, and hence there must exist an entropy current $S^{\mu}$ with non-negative divergence

$$e^{-1}\partial_{\mu}(eS^{\mu}) \geq 0 \,. \tag{3.1}$$

By the entropy current we mean the most general current, constructed from the fluid variables, up to first order in derivatives such that it reduces to $su^\mu$ for a perfect fluid and such that its divergence is non-negative for all fluid configurations. The requirement that the divergence of the entropy current is non-negative constrains the transport coefficients appearing in the expansion of the energy-momentum tensor. In this section, we elucidate the structure of the entropy current beyond perfect fluid order and, in particular, show that the appropriate fluid variables on curved space involve Lie derivatives of the geometric objects $\tau_\mu$ and $h_{\mu\nu}$ along $\beta^\mu = \frac{u^\mu}{T}$.

The *canonical* part of the entropy current is obtained by covariantizing (see e.g. [2]) the thermodynamic Euler relation (2.6)

$$s = \frac{1}{T}\tilde{\mathcal{E}} + \frac{1}{T}P \, , \tag{3.2}$$

leading to

$$S^\mu_{\text{can}} = -T^\mu{}_\nu \beta^\nu + P\beta^\mu \, , \tag{3.3}$$

such that $\tau_\mu S^\mu_{\text{can}} = s$ for a perfect fluid. However, there are generically terms present in the entropy current that do not arise in this way: such terms form the *non-canonical* piece of the entropy current, $S^\mu_{\text{non}}$, and we may in general write

$$S^\mu = S^\mu_{\text{can}} + S^\mu_{\text{non}} = -T^\mu{}_\nu \beta^\nu + P\beta^\mu + S^\mu_{\text{non}} \, . \tag{3.4}$$

Using the decomposition of the entropy current (3.4), we can recast the LHS of the second law (3.1) as

$$e^{-1}\partial_\mu(eS^\mu) = \left(T^\mu - T^\mu_{(0)}\right)\mathcal{L}_\beta \tau_\mu - \frac{1}{2}\left(T^{\mu\nu} - T^{\mu\nu}_{(0)}\right)\mathcal{L}_\beta h_{\mu\nu} + e^{-1}\partial_\mu\left(eS^\mu_{\text{non}}\right) \, , \tag{3.5}$$

where we used the Gibbs-Duhem relation (2.1) as well as

$$-T^\mu_{(0)}\mathcal{L}_\beta \tau_\mu + \frac{1}{2}T^{\mu\nu}_{(0)}\mathcal{L}_\beta h_{\mu\nu} = e^{-1}\partial_\mu(eP\beta^\mu) \, , \tag{3.6}$$

with the perfect fluid energy-momentum tensors $T^\mu_{(0)}$ and $T^{\mu\nu}_{(0)}$ given in Eqs. (2.25) and (2.26). Here, the subscript (0) indicates that these terms are zeroth order in derivatives and thus correspond to the perfect fluid contributions.

Following the classification in refs. [25, 26], we split the currents $T^\mu - T^\mu_{(0)}$ and $T^{\mu\nu} - T^{\mu\nu}_{(0)}$, which are at least first order in derivatives, into dissipative and non-dissipative parts. The latter are further subdivided into hydrostatic (HS) terms and non-hydrostatic (NHS) terms, to be defined shortly, so that we find

$$T^\mu - T^\mu_{(0)} \;=\; T^\mu_{\text{D}} + T^\mu_{\text{HS}} + T^\mu_{\text{NHS}} \, , \tag{3.7}$$

$$T^{\mu\nu} - T^{\mu\nu}_{(0)} \;=\; T^{\mu\nu}_{\text{D}} + T^{\mu\nu}_{\text{HS}} + T^{\mu\nu}_{\text{NHS}} \, . \tag{3.8}$$

We will now define these three contributions separately. They should be thought of as independent contributions to the energy-momentum tensor and they all have the same symmetry properties with respect to rotations (and boosts or scale symmetries if these are present). For example spatial rotational symmetries dictate that $T^{\mu\nu}_{\text{D}}$, $T^{\mu\nu}_{\text{HS}}$ and $T^{\mu\nu}_{\text{NHS}}$ are all separately symmetric under the interchange of $\mu$ and $\nu$.

The dissipative terms produce entropy,

$$e^{-1}\partial_\mu(eS^\mu) = T^\mu_{\text{D}}\mathcal{L}_\beta \tau_\mu - \frac{1}{2}T^{\mu\nu}_{\text{D}}\mathcal{L}_\beta h_{\mu\nu} \geq 0 \, , \tag{3.9}$$

with equality holding if and only if $T_{\mathrm{D}}^{\mu} = T_{\mathrm{D}}^{\mu\nu} = 0$, while the non-dissipative NHS terms by definition obey,

$$T_{\mathrm{NHS}}^{\mu}\mathcal{L}_{\beta}\tau_{\mu} - \frac{1}{2}T_{\mathrm{NHS}}^{\mu\nu}\mathcal{L}_{\beta}h_{\mu\nu} = 0 \ . \tag{3.10}$$

These terms thus make a vanishing contribution to the divergence of the canonical entropy current. Finally, the non-dissipative HS terms are defined to cancel the divergence of the non-canonical entropy current,

$$e^{-1}\partial_{\mu}\left(eS_{\mathrm{non}}^{\mu}\right) = -T_{\mathrm{HS}}^{\mu}\mathcal{L}_{\beta}\tau_{\mu} + \frac{1}{2}T_{\mathrm{HS}}^{\mu\nu}\mathcal{L}_{\beta}h_{\mu\nu} \ , \tag{3.11}$$

and will play a major role in the next section. Note that (3.10) implies that the HS energy-momentum tensor is only defined up the addition of NHS terms, since they will leave (3.11) invariant. This is analogous to what happens when solving inhomogeneous differential equations, where any solution of the homogeneous equation can be added to a particular solution of the inhomogeneous equation to obtain a new solution of the inhomogeneous equation. Because of this, there is an inherent ambiguity in HS transport. We will make a specific choice that fixes this ambiguity as will be detailed in Sec. 4, where we construct actions for HS and NHS transport respectively.

The goal of this work will be to classify the allowed terms appearing in the three parts: D, HS and NHS of the energy-momentum tensor. We will start this analysis with a detailed study of the HS and NHS terms for which it is possible to write down a Lagrangian.

## 4 Non-dissipative transport

In this section we study Lagrangian descriptions of non-dissipative transport. We will see in Section 5 and Appendix B, by looking at constitutive relations, that at first order in derivatives all non-dissipative transport coefficients can be obtained from an action. We start with the hydrostatic partition function for fluids in thermal equilibrium. This requires curved backgrounds admitting a Killing vector that generates time translations. We will then relax the condition that there is a Killing vector, moving away from thermal equilibrium. This leads to an action for HS transport. The relaxation of the presence of a Killing vector allows for more terms to be added to the action. These extra terms all correspond to NHS transport as we will show in the last subsection of this section.

### 4.1 Hydrostatic partition function

If we assume a stationary curved background $\mathcal{M}_S$ with a time-translation symmetry generated by $H$, we can write down the thermal partition function

$$\mathcal{Z} = \mathrm{Tr}\left[e^{-H/T}\right] \ , \tag{4.1}$$

which, provided the Aristotelian background curves sufficiently weakly compared to the mean free path, is known as the hydrostatic partition function or the equilibrium partition function [27,28] (see also [21] for the construction of the hydrostatic partition function in the context of Newton–Cartan backgrounds). The time-translation symmetry implies that (4.1) gives rise to static responses. Phrased in the language of geometry, we take the time-translation symmetry of the background, which is described by $\tau_{\mu}$ and $h_{\mu\nu}$, to be generated by a timelike Killing vector $\beta^{\mu}$, where $\beta^{\mu} = (1, 0, \ldots, 0)$ in suitable coordinates. Timelike (and future pointing)

means that $\tau_\mu \beta^\mu > 0$. The Killing equations for an Aristotelian geometry are[8]

$$\mathcal{L}_\beta \tau_\mu = 0, \tag{4.2}$$

$$\mathcal{L}_\beta h_{\mu\nu} = 0. \tag{4.3}$$

These conditions imply equilibrium via (2.36) as they trivially solve the leading order equations of motion. The vector $\beta^\mu$ leads to a preferred choice of local temperature and velocity given by

$$T = 1/(\tau_\mu \beta^\mu) \tag{4.4}$$

$$u^\mu = T\beta^\mu, \tag{4.5}$$

where the velocity $u^\mu$ satisfies $\tau_\mu u^\mu = 1$, cf. (2.20). We see that the requirement of positive temperature is equivalent to the requirement that $\beta^\mu$ is future-pointing timelike, i.e. $\tau_\mu \beta^\mu > 0$.

To make contact with the thermal partition function (4.1), we need to analytically continue 'time', which we identify with the affine parameter $\lambda_t$ along integral curves of $\beta^\mu$, $\lambda_t \to -i\lambda_t^{\mathrm{E}}$. We then compactify this to the 'thermal circle' by identifying $\lambda_t^{\mathrm{E}} \sim \lambda_t^{\mathrm{E}} + 1/T$. In this way, a functional integral over the Euclideanized manifold will return the partition function $\mathcal{Z}$ in (4.1). Now, $\mathcal{Z}$ itself can be written in a derivative expansion, and by writing

$$S_{\mathrm{HPF}} = -i \log \mathcal{Z}, \tag{4.6}$$

we can now also expand $S_{\mathrm{HPF}}$ in derivatives

$$S_{\mathrm{HPF}} = \sum_n S_{\mathrm{HPF}}^{(n)}, \tag{4.7}$$

where $S_{\mathrm{HPF}}^{(n)}$ takes the form of an integral over $\mathcal{M}_S$ built from objects with $n$ derivatives. With a slight abuse of terminology, we will refer to $S_{\mathrm{HPF}}$ as the hydrostatic partition function.

In order to construct the hydrostatic partition function explicitly, we need to identify the allowed terms up to first order in derivatives taking into consideration the conditions of thermal equilibrium on a curved background imposed by the Killing equation. The first Killing equation (4.2) can be written as

$$T^{-1}\partial_\mu T - u^\nu \left(\partial_\nu \tau_\mu - \partial_\mu \tau_\nu\right) = 0, \tag{4.8}$$

while the second Killing equation (4.3) can be written as

$$\mathcal{L}_u h_{\mu\nu} - \frac{u^\rho}{T} h_{\rho\nu} \partial_\mu T - \frac{u^\rho}{T} h_{\rho\mu} \partial_\nu T = 0. \tag{4.9}$$

By contracting (4.8) with $u^\mu$ and $v^\mu$ we obtain

$$0 = u^\mu \partial_\mu T, \tag{4.10}$$

$$0 = T^{-1} v^\mu \partial_\mu T + u^\mu \mathcal{L}_v \tau_\mu. \tag{4.11}$$

Contracting (4.9) with $v^\mu v^\nu$ gives nothing as a result of the fact that $h_{\mu\nu}$ has one zero eigenvalue with eigenvector $v^\mu$. Contracting with $u^\mu u^\nu$, $v^\mu u^\nu$ and $h^{\mu\nu}$ leads to

$$0 = u^\mu \partial_\mu u^2, \tag{4.12}$$

---

[8]The field $h_{\mu\nu}$ is constrained to have one zero eigenvalue, so one can write it as $h_{\mu\nu} = \delta_{ab} e_\mu^a e_\nu^b$ in terms of unconstrained spatial vielbeins. Since the latter transform under local rotations, the Killing vector equation (4.3) can equivalently be written as

$$\mathcal{L}_\beta e_\mu^a = \lambda^a{}_b e_\mu^b,$$

where $\lambda^a{}_b = -\lambda_b{}^a$ is an infinitesimal local rotation.

$$0 = \frac{1}{2}v^\mu\partial_\mu u^2 - \frac{1}{2}u^\mu u^\nu \mathcal{L}_v h_{\mu\nu} + u^2 u^\mu \mathcal{L}_v \tau_\mu, \tag{4.13}$$

$$0 = e^{-1}\partial_\mu(e u^\mu), \tag{4.14}$$

where we defined $u^2 = h_{\mu\nu}u^\mu u^\nu$.

In order to construct the hydrostatic partition function $S_{\text{HPF}}$, we write down the most general expansion in derivatives of the background fields $\tau_\mu$ and $h_{\mu\nu}$ under the assumption that the Killing equations (4.2) and (4.3) are obeyed. At zeroth order in derivatives, we can build two scalars,

$$T, \quad u^2. \tag{4.15}$$

At first order, taking into account the relations (4.10)–(4.14), there are two independent one-derivative scalars, which we can take to be

$$v^\mu\partial_\mu T, \qquad v^\mu\partial_\mu u^2. \tag{4.16}$$

Scalars such as $u^\mu\partial_\mu T$, $u^\mu\partial_\mu u^2$, $e^{-1}\partial_\mu(e u^\mu)$, $u^\mu \mathcal{L}_v \tau_\mu$, $u^\mu u^\nu \mathcal{L}_v h_{\mu\nu}$ are either zero or related to (4.16) via the relations (4.10)–(4.14), possibly using partial integration. Furthermore, scalars such as $h_{\mu\nu}u^\mu \mathcal{L}_v u^\nu$, $h^{\mu\nu}\mathcal{L}_u h_{\mu\nu}$ and $h^{\mu\nu}\mathcal{L}_v h_{\mu\nu}$ do not lead to anything new as they can be rewritten in terms of (4.16). Up to first order, we therefore obtain

$$S_{\text{HPF}} = \int d^{d+1}x\, e \left( P(T, u^2) + F_1(T, u^2)v^\mu\partial_\mu T + F_2(T, u^2)v^\mu\partial_\mu u^2 \right) + \mathcal{O}(\partial^2), \tag{4.17}$$

where the functions $P$, $F_1$ and $F_2$ are all arbitrary functions of $T$ and $u^2$.

Since the background is stationary we can use adapted coordinates (known as 'static gauge' in [21]), where we choose a time direction $t$ such that the Killing vector $\beta^\mu$ is given by $\beta^\mu = \delta_t^\mu$. As $\beta^\mu$ is Killing, the tensors $\tau_\mu$ and $h_{\mu\nu}$ are independent of $t$. Thus, in these coordinates the fluid velocity $u^\mu$ is given by[9] $u^\mu = T(x)\delta_t^\mu$, where the temperature $T$ only depends on $x^i$ with $i = 1,\dots,d$ and not on $t$.

The Euclideanized background has the structure of a fiber bundle, where the thermal circle is fibered over the spatial base (see also [21]). We can then perform a timelike Kaluza–Klein reduction of our Aristotelian geometry to arrive at $S_{\text{HPF}}$ in terms of fields that are unconstrained by the Killing equations. In particular, in our adapted coordinates $\tau_\mu$ and $h_{\mu\nu}$ can be parameterized as

$$\tau_\mu dx^\mu = N(dt - A_i dx^i), \tag{4.18}$$

$$h_{\mu\nu}dx^\mu dx^\nu = \sigma_{ij}\left(dx^i + X^i(dt - A)\right)\left(dx^j + X^j(dt - A)\right). \tag{4.19}$$

The metric $\sigma_{ij}$ is invertible and has signature $(1,\dots,1)$. This parameterization makes manifest that $h_{\mu\nu}$ has one zero eigenvalue. The integration measure $e$ is $e = N\sqrt{\sigma}$ where $\sigma = \det\sigma_{ij}$. Further, the 1-form $A = A_i dx^i$ is a Kaluza–Klein type gauge connection in that $\delta t = \Lambda(x)$ and $\delta A_i = \partial_i\Lambda$ leave the parameterization invariant. The other fields $N$, $X^i$ and $\sigma_{ij}$, which depend on $x^i$ but not on $t$, are thus all gauge invariant. Since $\tau_\mu u^\mu = 1$ with $u^\mu = T(x)\delta_t^\mu$ it must be that $T = N^{-1}$. Finally, the vector $v^\mu$ satisfying $\tau_\mu v^\mu = -1$ is given by

$$v^\mu = -N^{-1}\delta_t^\mu + N^{-1}X^i\left(\delta_i^\mu + A_i\delta_t^\mu\right). \tag{4.20}$$

We can now ask again what are the invariant scalars up to first order in derivatives. These have to be gauge invariant under the Kaluza–Klein gauge transformation $\delta A_i = \partial_i\Lambda$. At zeroth

---

[9]Note that since $u^\mu \propto \delta_t^\mu$, we are in a comoving frame.

order in derivatives, we find $N$ and $X^2 = \sigma_{ij} X^i X^j$, so that at first order in derivatives we can build two scalars

$$X^i \partial_i N\,, \qquad X^i \partial_i X^2\,. \tag{4.21}$$

We thus obtain the hydrostatic partition function

$$S = \int_{\Sigma} d^d x \sqrt{\sigma} \left( \tilde{P}(N, X^2) + G_1(N, X^2) X^i \partial_i N + G_2(N, X^2) X^i \partial_i X^2 \right)\,. \tag{4.22}$$

A term such as $G_3(N, X^2) \partial_i \left( \sqrt{\sigma} X^i \right)$ can be absorbed into the $G_1$ and $G_2$ terms after partial integration. Likewise, a term such as $G_4(N, X^2) \sqrt{\sigma} \sigma^{ij} \mathcal{L}_X \sigma_{ij}$, where $\sigma^{ij}$ is the inverse of $\sigma_{ij}$, can be written as $2 G_4(N, X^2) \partial_i \left( \sqrt{\sigma} X^i \right)$ so that this, too, is nothing new. Thus we see that this line of reasoning leads to the same hydrostatic partition function as in (4.17). For ease of comparing (4.22) and (4.17) we note that the vanishing of $u^\mu \partial_\mu T$, $u^\mu \partial_\mu u^2$ and $\partial_\mu (eu^\mu)$ follows immediately from the $t$-independence of the fields involved in the Kaluza–Klein reduction. Furthermore one observes that $e v^\mu \partial_\mu F = \sqrt{\sigma} X^i \partial_i F$ where $F$ is any function of $T = N^{-1}$ and $u^2 = N^{-2} X^2$. We can now set (4.22) and (4.17) equal to each other and in principle read off the 1-1 relation between the sets of functions $\{P, F_1, F_2\}$ and $\{\tilde{P}, G_1, G_2\}$.

## 4.2 Action for hydrostatic non-dissipative transport

In order to compute transport coefficients using (4.17), we will drop the restriction to stationary configurations – that is to say, we relax the requirement that $\beta^\mu$ is Killing. This will lead to an action for the hydrostatic non-dissipative transport coefficients[10] This is related to the discussion below equation (3.7) in the following way. As we will show, the energy-momentum tensor obtained by varying the geometric variables in the action that follows from the hydrostatic partition function without the condition that $\beta^\mu$ is Killing, is equal to the HS part of the energy-momentum tensor as defined in equation (3.11).

We now have geometric variables $\tau_\mu$ and $h_{\mu\nu}$ and fluid variables $\beta^\mu$. However as shown in [26] we cannot freely vary $\beta^\mu$. Instead we think of it as being described in terms of fundamental variables whose variation is such that we vary $\beta^\mu$ under a diffeomorphism. We thus obtain the fluid equations of motion via diffeomorphism invariance, i.e.

$$\delta_\xi S_{\text{HS}} = \int_{\mathcal{M}} d^{d+1} x \, e \left( -T^\mu \delta_\xi \tau_\mu + \frac{1}{2} T^{\mu\nu} \delta_\xi h_{\mu\nu} + F_\mu \delta_\xi \beta^\mu \right) = 0\,. \tag{4.23}$$

Here $S_{\text{HS}}$ is the same action as in (4.17) except that now $\beta^\mu$ is no longer a Killing vector, and $F_\mu$ is the response to varying $\beta^\mu$ under diffeomorphisms. Setting the diffeomorphism variation to zero for any $\xi^\mu$ leads to the off-shell diffeomorphism Ward identity,

$$e^{-1} \partial_\mu \left( e T^\mu{}_\rho \right) + T^\mu \partial_\rho \tau_\mu - \frac{1}{2} T^{\mu\nu} \partial_\rho h_{\mu\nu} = F_\mu \partial_\rho \beta^\mu + e^{-1} \partial_\mu (e F_\rho \beta^\mu)\,, \tag{4.24}$$

where we recall $T^\mu{}_\nu = -T^\mu \tau_\nu + T^{\mu\rho} h_{\rho\nu}$. Using that the fluid equations of motion follow from a diffeomorphism transformation of $\beta^\mu$ we see that on shell the left- and right-hand side vanish separately. We conclude that in order to compute the energy-momentum tensor we vary $\tau_\mu$ and $h_{\mu\nu}$ keeping $\beta^\mu$ fixed, and furthermore that the on-shell energy-momentum conservation equation is given by

$$e^{-1} \partial_\mu \left( e T^\mu{}_\rho \right) + T^\mu \partial_\rho \tau_\mu - \frac{1}{2} T^{\mu\nu} \partial_\rho h_{\mu\nu} = 0\,, \tag{4.25}$$

---

[10]In terms of the classification of [25, 26], the non-dissipative transport coefficients considered in this section are class $L = H_S \cup \bar{H}_S$, i.e. those that have a Lagrangian description.

as stated before in Eq. (2.27).

We now first show that we can reproduce the perfect fluid equations of motion on an arbitrary curved background as discussed in Section 2.3. To this end we consider the action up to zeroth order in derivatives, i.e.

$$S_{(0)} = \int_{\mathcal{M}} d^{d+1}x \; e P(T, u^2),$$
(4.26)

with $P$ the pressure as we will see a posteriori. Since we vary the background sources keeping $\beta^\mu$ fixed, we have $\delta T = -T u^\mu \delta \tau_\mu$ and $\delta u^\mu = -u^\mu u^\rho \delta \tau_\rho$. Using further that $\delta e = e\left(-v^\mu \delta \tau_\mu + \frac{1}{2} h^{\mu\nu} \delta h_{\mu\nu}\right)$. We then find

$$T_{(0)}^\mu = P v^\mu + \left(\frac{\partial P}{\partial T}\right)_{u^2} T u^\mu + 2\left(\frac{\partial P}{\partial u^2}\right)_T u^2 u^\mu,$$
(4.27)

$$T_{(0)}^{\mu\nu} = P h^{\mu\nu} + 2\left(\frac{\partial P}{\partial u^2}\right)_T u^\mu u^\nu.$$
(4.28)

Using the thermodynamic relations $\left(\frac{\partial P}{\partial T}\right)_{u^2} = s$, $\left(\frac{\partial P}{\partial u^2}\right)_T = \frac{1}{2}\rho$, $sT = \tilde{\mathcal{E}} + P$ as well as the relation (2.21) between $v^\mu$ and $u^\mu$, we recover the perfect fluid energy-momentum tensor (2.25) and (2.26).

Let us next consider the first order derivative terms in (4.17). We will denote the first order part of the action by $S_{(1)}$, i.e.

$$S_{(1)} = \int d^{d+1}x \, e\left(F_1(T, u^2) v^\mu \partial_\mu T + F_2(T, u^2) v^\mu \partial_\mu u^2\right).$$
(4.29)

We thus have $S_{\mathrm{HS}} = S_{(0)} + S_{(1)}$.

It is well known that when we introduce derivative corrections, the notion of temperature and velocity can undergo field redefinitions whereby two equally valid definitions of temperature and velocity can differ by derivatives of the fluid variables. Such redefinitions are known as hydrodynamical frame transformations and choosing a certain set of fluid variables corresponds to choosing a hydro frame. We will present our final results in Landau frame, which is defined by declaring that the full (all order in derivatives) energy-momentum tensor is such that

$$T^\mu{}_\nu u^\nu = -\tilde{\mathcal{E}} u^\mu,$$
(4.30)

where $-\tilde{\mathcal{E}}$ is the unique negative eigenvalue of the energy-momentum tensor which is taken to be equal to its perfect fluid value. The corresponding eigenvector $u^\mu$ is used to define the velocity. This equation does not fix the normalization of $u^\mu$. The choice of eigenvalue and of eigenvector (up to rescaling) are thus $d+1$ conditions that can be used to fix the definition of $T$ and $u^\mu$. As in (2.20), we will choose the normalization $\tau_\mu u^\mu = 1$.

When including the first order derivatives in the action $S_{\mathrm{HS}}$ and computing the energy-momentum tensor by variation, we do not end up with a Landau frame expression. We will refer to the frame in which $S_{\mathrm{HS}}$ is written as the Lagrangian frame, and to indicate this frame dependence we will write the variation of $S_{(1)}$ as

$$\delta S_{(1)} = \int_{\mathcal{M}} d^{d+1}x \; e\left(-\mathcal{T}_{(1)}^\mu \delta \tau_\mu + \frac{1}{2}\mathcal{T}_{(1)}^{\mu\nu} \delta h_{\mu\nu}\right),$$
(4.31)

i.e. we denote the responses with calligraphic $\mathcal{T}$. The total energy-momentum tensor must be frame-independent, so we have the equation

$$\mathcal{T}_{(0)}^\mu + \mathcal{T}_{(1)}^\mu = T_{(0)}^\mu + T_{(1)\mathrm{HS}}^\mu,$$
(4.32)

$$\mathcal{T}^{\mu\nu}_{(0)} + \mathcal{T}^{\mu\nu}_{(1)} = T^{\mu\nu}_{(0)} + T^{\mu\nu}_{(1)\text{HS}}, \tag{4.33}$$

where the left hand side is in Lagrangian frame and is computed by variation of the action, while the right-hand side is in any frame – for example in Landau frame. The right hand side is computed by applying a frame transformation to the left hand side and arranging the result according to the number of derivatives. At perfect fluid order the expressions look the same, but they are written with respect to different choices of $T$ and $u^\mu$.

Let us next compute the variation of the first derivative terms in the action (4.29). Using $\delta v^\lambda = v^\lambda v^\mu \delta\tau_\mu - h^{\lambda(\mu} v^{\nu)} \delta h_{\mu\nu}$, we obtain

$$\mathcal{T}^{\mu}_{(1)} = u^\mu \left[ \left( \frac{\partial F_1}{\partial u^2} \right)_T - \left( \frac{\partial F_2}{\partial T} \right)_{u^2} \right] (2u^2 v^\lambda \partial_\lambda T - Tv^\lambda \partial_\lambda u^2) + u^\mu K(TF_1 + 2u^2 F_2), \tag{4.34}$$

$$\mathcal{T}^{\mu\nu}_{(1)} = \left( h^{\mu\nu} v^\lambda - h^{\lambda\mu} v^\nu - h^{\lambda\nu} v^\mu \right) \left( F_1 \partial_\lambda T + F_2 \partial_\lambda u^2 \right)$$
$$+ 2 \left[ \left( \frac{\partial F_1}{\partial u^2} \right)_T - \left( \frac{\partial F_2}{\partial T} \right)_{u^2} \right] u^\mu u^\nu v^\lambda \partial_\lambda T + 2F_2 K u^\mu u^\nu, \tag{4.35}$$

where $K$ is the trace of the extrinsic curvature defined in equation (2.15). Combining these according to $\mathcal{T}^\mu_{(1)\nu} = -\mathcal{T}^\mu_{(1)} \tau_\nu + \mathcal{T}^{\mu\rho}_{(1)} h_{\rho\nu}$ yields the first order part of the energy-momentum tensor in Lagrangian frame,

$$\mathcal{T}^\mu_{(1)\nu} = \left[ 2v^\lambda u^\mu h_{\sigma\rho} \Pi^\sigma{}_\nu u^\rho \left[ \left( \frac{\partial F_1}{\partial u^2} \right)_T - \left( \frac{\partial F_2}{\partial T} \right)_{u^2} \right] + 2F_1 v^{[\lambda} h^{\mu]\rho} h_{\rho\nu} \right] \partial_\lambda T$$
$$+ \left[ v^\lambda u^\mu \tau_\nu T \left[ \left( \frac{\partial F_1}{\partial u^2} \right)_T - \left( \frac{\partial F_2}{\partial T} \right)_{u^2} \right] + 2F_2 v^{[\lambda} h^{\mu]\rho} h_{\rho\nu} \right] \partial_\lambda u^2$$
$$- Tu^\mu \tau_\nu F_1 K + 2F_2 K u^\mu \Pi^\sigma{}_\nu h_{\sigma\rho} u^\rho. \tag{4.36}$$

This should be added to the perfect fluid energy-momentum tensor

$$\mathcal{T}^\mu_{(0)\nu} = -\left( \tilde{\mathcal{E}} + P + \rho u^2 \right) u^\mu \tau_\nu + \rho u^\mu u^\rho h_{\rho\nu} + P\delta^\mu_\nu, \tag{4.37}$$

coming from the variation of $S_{(0)}$.

In Appendix A we work out the transformation from any frame to Landau frame, indicated by primed variables. Using the results from that appendix we obtain in Landau frame that

$$T^\mu_{(1)\text{HS}\nu} = T^{\mu\rho}_{(1)\text{HS}} h_{\rho\sigma} \Pi'^\sigma{}_\nu, \tag{4.38}$$

where we remind the reader that $T^{\mu\rho}_{(1)\text{HS}}$ is computed using (4.33). This gives

$$T^{\mu\rho}_{(1)\text{HS}} = \mathcal{T}^\kappa_{(1)\lambda} \Pi'^\sigma{}_\kappa \frac{u'^\lambda}{s'T'} \left[ \rho' \left( u'^\mu \delta^\rho_\sigma + u'^\rho \delta^\mu_\sigma \right) + 2u'_\sigma \left( h^{\mu\rho} \left( \frac{\partial P'}{\partial u'^2} \right)_{s'} + u'^\mu u'^\rho \left( \frac{\partial \rho'}{\partial u'^2} \right)_{s'} \right) \right]$$
$$+ \mathcal{T}^{\mu\rho}_{(1)} + \mathcal{T}^\sigma_{(1)\nu} \tau_\sigma \frac{u'^\nu}{T'} \left[ u'^\mu u'^\rho \left( \frac{\partial \rho'}{\partial s'} \right)_{u'^2} + h^{\mu\rho} \left( \frac{\partial P'}{\partial s'} \right)_{u'^2} \right], \tag{4.39}$$

with the prime denoting Landau frame fluid variables. We defined $u'_\sigma = h_{\sigma\kappa} u'^\kappa$. Terms such as $\mathcal{T}^{\mu\rho}_{(1)}$ are given in (4.35), but where we must replace the $T$ and $u^\mu$ by $T'$ and $u'^\mu$.

We will drop the primes and use the relations

$$\partial_\mu T = -T^2 \mathcal{L}_\beta \tau_\mu - Tu^\rho \tau_{\mu\rho}, \tag{4.40}$$

$$\partial_\mu u^2 = Tu^\nu \left( \mathcal{L}_\beta h_{\mu\nu} - u_\mu \mathcal{L}_\beta \tau_\nu - u_\nu \mathcal{L}_\beta \tau_\mu \right) - u^2 u^\rho \tau_{\mu\rho} - u^\rho \omega_{\rho\mu}, \tag{4.41}$$

where $\tau_{\mu\nu}$ is the torsion 2-form defined in (2.18) and where $\omega_{\rho\mu} = \partial_\rho u_\mu - \partial_\mu u_\rho$. Using the equations of motion (2.37) to eliminate $\mathcal{L}_\beta \tau_\mu$ derivatives, allows us to write

$$T^{\mu\nu}_{(1)\text{HS}} = \frac{1}{2} \eta^{\mu\nu\alpha\beta}_{\text{HS}} \left( \mathcal{L}_\beta h_{\alpha\beta} - u_\alpha \mathcal{L}_\beta \tau_\beta - u_\beta \mathcal{L}_\beta \tau_\alpha \right) + \frac{1}{2} \eta^{\mu\nu\alpha\beta}_{\text{tor}} \tau_{\alpha\beta}$$

$$+\frac{1}{2}\eta_{\text{rot}}^{\mu\nu\alpha\beta}\omega_{\alpha\beta}+\frac{1}{2}\eta_{\text{ext}}^{\mu\nu\alpha\beta}K_{\alpha\beta}\,, \tag{4.42}$$

where $K_{\alpha\beta}$ is the extrinsic curvature introduced in (2.13). In obtaining this result we have also used the following relation

$$
\begin{aligned}
\nu^{\mu}\omega_{\mu\nu}=&\ T\nu^{\mu}\left(\mathcal{L}_{\beta}h_{\mu\nu}-u_{\mu}\mathcal{L}_{\beta}\tau_{\nu}-u_{\nu}\mathcal{L}_{\beta}\tau_{\mu}\right)\\
&-u_{\nu}u^{\rho}\nu^{\mu}\tau_{\mu\rho}-2u^{\rho}K_{\nu\rho}\,.
\end{aligned}
\tag{4.43}
$$

This was done in order to make $\eta_{\text{rot}}^{\mu\nu\alpha\beta}=-\eta_{\text{rot}}^{\mu\nu\beta\alpha}$ spatial in its last two indices (to avoid ambiguities among some of the $\eta$ tensors as a result of (4.43)), i.e. $\tau_{\alpha}\eta_{\text{rot}}^{\mu\nu\alpha\beta}=0$.

The $\eta_{\text{HS}}^{\mu\nu\alpha\beta}$ tensor that features in (4.42) can be written as[11]

$$
\begin{aligned}
\eta_{\text{HS}}^{\mu\nu\alpha\beta}=&\ \mathcal{J}_{1}h^{\mu\nu}h^{\alpha\beta}+\mathcal{J}_{2}u^{\mu}u^{\nu}u^{\alpha}u^{\beta}+4\mathcal{J}_{3}\nu^{(\mu}h^{\nu)(\alpha}\nu^{\beta)}+\frac{1}{2}\mathcal{J}_{4}(h^{\mu\nu}u^{\alpha}u^{\beta}+h^{\alpha\beta}u^{\mu}u^{\nu})\\
&+\mathcal{J}_{5}(h^{\mu\nu}u^{(\alpha}\nu^{\beta)}+h^{\alpha\beta}u^{(\mu}\nu^{\nu)})+\mathcal{J}_{6}(u^{\mu}u^{\nu}u^{(\alpha}\nu^{\beta)}+u^{\alpha}u^{\beta}u^{(\mu}\nu^{\nu)})\\
&+2\mathcal{J}_{7}(\nu^{(\mu}h^{\nu)(\alpha}u^{\beta)}+\nu^{(\alpha}h^{\beta)(\mu}u^{\nu)})+\frac{1}{2}\mathcal{A}_{1}(h^{\mu\nu}u^{\alpha}u^{\beta}-h^{\alpha\beta}u^{\mu}u^{\nu})\\
&+\mathcal{A}_{2}(h^{\mu\nu}u^{(\alpha}\nu^{\beta)}-h^{\alpha\beta}u^{(\mu}\nu^{\nu)})+\mathcal{A}_{3}(u^{\mu}u^{\nu}u^{(\alpha}\nu^{\beta)}-u^{\alpha}u^{\beta}u^{(\mu}\nu^{\nu)})\\
&+2\mathcal{A}_{4}(\nu^{(\mu}h^{\nu)(\alpha}u^{\beta)}-\nu^{(\alpha}h^{\beta)(\mu}u^{\nu)})\,,
\end{aligned}
\tag{4.44}
$$

where the eleven scalars $\mathcal{J}_{1,\dots,7},\mathcal{A}_{1,2,3,4}$ are given by

$$\mathcal{J}_{1}=\frac{T}{s}F_{1}\left[s\left(\frac{\partial P}{\partial s}\right)_{u^{2}}-2u^{2}\left(\frac{\partial P}{\partial u^{2}}\right)_{s}\right]\,, \tag{4.45}$$

$$
\begin{aligned}
\mathcal{J}_{2}=&\ \frac{2}{s}F_{2}\left[s\left(\frac{\partial\rho}{\partial s}\right)_{u^{2}}-2u^{2}\left(\frac{\partial\rho}{\partial u^{2}}\right)_{s}-2\rho\right]+\frac{4u^{2}}{s}f_{A}\left(\frac{\partial\rho}{\partial u^{2}}\right)_{s}\\
&-\frac{2}{s}f_{B}\left[s\left(\frac{\partial\rho}{\partial s}\right)_{u^{2}}-2\rho\right]\,,
\end{aligned}
\tag{4.46}
$$

$$\mathcal{J}_{3}=-TF_{2}\,, \tag{4.47}$$

$$
\begin{aligned}
\mathcal{J}_{4}=&\ \frac{T}{s}F_{1}\left[s\left(\frac{\partial\rho}{\partial s}\right)_{u^{2}}-2u^{2}\left(\frac{\partial\rho}{\partial u^{2}}\right)_{s}-2\rho\right]\,,\\
&+\frac{2}{s}(F_{2}-f_{B})\left[s\left(\frac{\partial P}{\partial s}\right)_{u^{2}}-2u^{2}\left(\frac{\partial P}{\partial u^{2}}\right)_{s}\right]\,,
\end{aligned}
\tag{4.48}
$$

$$\mathcal{J}_{5}=\frac{T}{s}F_{1}\left[2\left(\frac{\partial P}{\partial u^{2}}\right)_{s}-\rho\right]+2F_{2}\left[\left(\frac{\partial P}{\partial s}\right)_{u^{2}}+T\right]-2f_{A}\left(\frac{\partial P}{\partial s}\right)_{u^{2}}\,, \tag{4.49}$$

$$\mathcal{J}_{6}=\frac{2T}{s}F_{1}\left(\frac{\partial\rho}{\partial u^{2}}\right)_{s}+\frac{2}{s}F_{2}\left[s\left(\frac{\partial\rho}{\partial s}\right)_{u^{2}}-\rho\right]-2f_{A}\left(\frac{\partial\rho}{\partial s}\right)_{u^{2}}+\frac{2\rho}{s}f_{B}\,, \tag{4.50}$$

$$\mathcal{J}_{7}=\frac{\rho T}{s}F_{1}-TF_{2}\,, \tag{4.51}$$

$$
\begin{aligned}
\mathcal{A}_{1}=&\ -\frac{4\rho}{s}F_{1}\left(\frac{\partial P}{\partial s}\right)_{u^{2}}+\frac{TF_{1}}{s}\left[s\left(\frac{\partial\rho}{\partial s}\right)_{u^{2}}-2u^{2}\left(\frac{\partial\rho}{\partial u^{2}}\right)_{s}-2\rho\right]\\
&+\frac{4u^{2}}{s}F_{1}\left[\left(\frac{\partial P}{\partial u^{2}}\right)_{s}\left(\frac{\partial\rho}{\partial s}\right)_{u^{2}}-\left(\frac{\partial\rho}{\partial u^{2}}\right)_{s}\left(\frac{\partial P}{\partial s}\right)_{u^{2}}\right]\\
&+\frac{2}{s}(F_{2}+f_{B})\left[s\left(\frac{\partial P}{\partial s}\right)_{u^{2}}-2u^{2}\left(\frac{\partial P}{\partial u^{2}}\right)_{s}\right]\,,
\end{aligned}
\tag{4.52}
$$

$$\mathcal{A}_{2}=-\frac{F_{1}}{s}\left[\rho T+2\rho\left(\frac{\partial P}{\partial s}\right)_{u^{2}}+2T\left(\frac{\partial P}{\partial u^{2}}\right)_{s}\right]+2F_{2}\left[\left(\frac{\partial P}{\partial s}\right)_{u^{2}}+T\right]$$

---

[11]Note that $T_{(1)\text{HS}}^{\mu\nu}$ in (4.42) is only defined up to terms proportional to $\nu^{\mu}\nu^{\nu}$, since $\nu^{\mu}\nu^{\nu}\delta h_{\mu\nu}=0$.

$$-2f_A\left(\frac{\partial P}{\partial s}\right)_{u^2},\tag{4.53}$$

$$\mathcal{A}_3 = -2\frac{F_1}{s}\left[\rho\left(\frac{\partial \rho}{\partial s}\right)_{u^2}+T\left(\frac{\partial \rho}{\partial u^2}\right)_s\right]+2F_2\left(\frac{\partial \rho}{\partial s}\right)_{u^2}+\frac{2\rho}{s}F_2-2f_A\left(\frac{\partial \rho}{\partial s}\right)_{u^2}$$

$$+\frac{2\rho}{s}f_B,\tag{4.54}$$

$$\mathcal{A}_4 = \mathcal{J}_7,\tag{4.55}$$

where we defined the recurring combinations

$$f_A := T\left[\left(\frac{\partial F_2}{\partial T}\right)_{u^2}-\left(\frac{\partial F_1}{\partial u^2}\right)_T\right], \quad f_B := T\left[\left(\frac{\partial F_2}{\partial T}\right)_{u^2}-\left(\frac{\partial F_1}{\partial T}\right)_{u^2}\right].\tag{4.56}$$

The coefficients $\mathcal{J}_i$ make up the symmetric part of $\eta_{\mathrm{HS}}^{\mu\nu\alpha\beta}$ under the interchange of $\mu\nu$ and $\alpha\beta$ while the coefficients $\mathcal{A}_i$ make up the anti-symmetric part of $\eta_{\mathrm{HS}}^{\mu\nu\alpha\beta}$. The remaining $\eta$ tensors in (4.42) are given by

$$\eta_{\mathrm{tor}}^{\mu\nu\alpha\beta} = 2\left[\frac{1}{T}\left(TF_1+2u^2F_2\right)\left[\left(\frac{\partial P}{\partial s}\right)_{u^2}+T\right]-\frac{2u^2}{T}\left(\frac{\partial P}{\partial s}\right)_{u^2}f_A\right]h^{\mu\nu}u^{[\alpha}v^{\beta]}$$

$$+\frac{2}{T}\left[\left(TF_1+2u^2F_2\right)\left(\frac{\partial \rho}{\partial s}\right)_{u^2}-2u^2\left(\frac{\partial \rho}{\partial s}\right)_{u^2}f_A-2Tf_B\right]u^\mu u^\nu u^{[\alpha}v^{\beta]}$$

$$+4(TF_1+u^2F_2)v^{(\mu}h^{\nu)[\alpha}u^{\beta]}-4F_2v^{(\mu}u^{\nu)}u^{[\alpha}v^{\beta]}\tag{4.57}$$

$$\eta_{\mathrm{rot}}^{\mu\nu\alpha\beta} = -4F_2v^{(\mu}h^{\nu)[\alpha}h^{\beta]\sigma}h_{\sigma\kappa}u^\kappa,\tag{4.58}$$

$$\eta_{\mathrm{ext}}^{\mu\nu\alpha\beta} = -2\left[F_1\left(\frac{\partial \rho}{\partial s}\right)_{u^2}-2F_2\right]u^\mu u^\nu h^{\alpha\beta}-2F_1\left(\frac{\partial P}{\partial s}\right)_{u^2}h^{\mu\nu}h^{\alpha\beta}+8F_2v^{(\mu}h^{\nu)(\alpha}u^{\beta)}$$

$$-\frac{4}{T}\left[F_2\left[\left(\frac{\partial P}{\partial s}\right)_{u^2}+T\right]-f_A\left(\frac{\partial P}{\partial s}\right)_{u^2}\right]h^{\mu\nu}u^\alpha u^\beta$$

$$-\frac{4}{T}(F_2-f_A)\left(\frac{\partial \rho}{\partial s}\right)_{u^2}u^\mu u^\nu u^\alpha u^\beta.\tag{4.59}$$

An important consistency check for these results is performed in Section 5.6, where we show that these expressions recover the results of [11] in the limit of linearized perturbations around a fluid at rest. Equation (4.42) is the main result of this subsection. We will next discuss how this is related to the non-canonical entropy current as it should via the frame-independent definition (3.11).

## 4.3 Non-canonical entropy current

In the previous subsection, we obtained explicit expressions for the contributions to the energy-momentum tensor that arise from the action $S_{\mathrm{HS}}$. As discussed in Section 3, the HS part of the energy-momentum tensor is related to the divergence of the non-canonical part of the entropy current, cf. (3.11). The goal of this subsection is to show that there exists a non-canonical entropy current whose divergence obeys (3.11) where the energy-momentum tensor is the one we just obtained. The result for the non-canonical entropy current is given in equation (4.72) where we used the definitions (4.68) and (4.69).

In Appendix B we show that the converse is also true, i.e. starting from the most general non-canonical entropy current and demanding that its divergence obeys (3.11), where the energy-momentum tensor is the most general one allowed by symmetries, we find (using only on-shell relations) that the non-canonical entropy current is (up to terms that are identically conserved) precisely of the form as given in (4.70) and (4.72). The analysis in Appendix B has been restricted to flat space but we expect the result to generalize to any curved background.

In the Lagrangian frame, the divergence of the non-canonical entropy current (3.11) must obey

$$e^{-1}\partial_\mu\left(eS^\mu_{(1)\text{non}}\right) = -\mathcal{T}^\mu_{(1)}\mathcal{L}_\beta\tau_\mu + \frac{1}{2}\mathcal{T}^{\mu\nu}_{(1)}\mathcal{L}_\beta h_{\mu\nu}, \tag{4.60}$$

where $\mathcal{T}^\mu_{(1)}$ and $\mathcal{T}^{\mu\nu}_{(1)}$ are given in (4.34) and (4.35). This can be solved for $S^\mu_{(1)\text{non}}$ up to identically conserved currents, leading to

$$S^\mu_{(1)\text{non}} = -\frac{1}{T}v^\mu F_1 u^\rho\partial_\rho T + \frac{1}{T}u^\mu F_1 v^\rho\partial_\rho T - \frac{1}{T}v^\mu F_2 u^\rho\partial_\rho u^2 + \frac{1}{T}u^\mu F_2 v^\rho\partial_\lambda u^2. \tag{4.61}$$

The total entropy current (3.4) in the Lagrangian frame can be written as $S^\mu = su^\mu - \mathcal{T}^\mu_{(1)\nu}\beta^\nu + S^\mu_{(1)\text{non}}$ where we have split the contributions from the zeroth order derivative terms, which is just the perfect fluid result $su^\mu$, from the terms containing first order derivatives. Substituting the result obtained in the previous subsection for $\mathcal{T}^\mu_{(1)\nu}$ (see equation (4.36)) and using (4.61) we obtain for the full entropy current in Lagrangian frame,

$$S^\mu = su^\mu - u^\mu F_1 e^{-1}\partial_\rho(ev^\rho) + u^\mu\left(\frac{\partial F_2}{\partial T} - \frac{\partial F_1}{\partial u^2}\right)v^\rho\partial_\rho u^2. \tag{4.62}$$

One may at this point object that the full entropy current could also receive contributions from the dissipative sector of transport. We will show next that in the Lagrangian frame only the HS sector contributes to the entropy current.

To show this we first observe that in Landau frame (denoted here by a prime just like in Apppendix A) the total entropy current (3.4) is given by

$$S^\mu = s'u'^\mu + S^\mu_{(1)\text{non}}, \tag{4.63}$$

simply because Landau frame is equivalent to demanding that the canonical entropy current is that of a perfect fluid, i.e. $T^\mu_{(1)\nu}\beta^\nu \sim T^\mu_{(1)\nu}u^\nu = 0$ by definition. If we next take the Lagrangian frame result (4.62) and we transform it to Landau frame using (A.8) and (A.9) we obtain (4.63) with $S^\mu_{(1)\text{non}}$ as given in (4.61) (written in terms of primed variables). In other words the Lagrangian frame entropy current is the same as the total entropy current (4.63) and so since the total entropy current is frame independent it must be that (4.62) equals the total entropy current.

In Landau frame the right hand side of equation (3.11) can be written as

$$e^{-1}\partial_\mu\left(eS^\mu_{(1)\text{non}}\right) = \frac{1}{2T}T^{\mu\nu}_{(1)\text{HS}}\left(\mathcal{L}_u h_{\mu\nu} - u_\nu\mathcal{L}_u\tau_\mu - u_\mu\mathcal{L}_u\tau_\nu\right), \tag{4.64}$$

where $u_\mu = h_{\mu\nu}u^\nu$ and where $T^{\mu\nu}_{(1)\text{HS}}$ is the Landau frame expression for the HS contributions to $T^{\mu\nu}_{(1)}$. This was computed in the previous subsection in (4.39). As a consistency check we will explicitly verify that this is indeed the case.

To first order in derivatives, we can rewrite the right hand side of (4.60) in terms of $u'$ rather than $u$, and using the relation

$$-\mathcal{T}^\mu_{(1)} = \mathcal{T}^\mu_{(1)\nu}u'^\nu - \mathcal{T}^{\mu\rho}_{(1)}h_{\rho\nu}u'^\nu, \tag{4.65}$$

we find that

$$e^{-1}\partial_\mu\left(eS^\mu_{(1)\text{non}}\right) = \mathcal{T}^\mu_{(1)\rho}u'^\rho\mathcal{L}_{\beta'}\tau_\mu + \frac{1}{2}\mathcal{T}^{\mu\nu}_{(1)}\left(\mathcal{L}_{\beta'}h_{\mu\nu} - h_{\nu\rho}u'^\rho\mathcal{L}_{\beta'}\tau_\mu - h_{\mu\rho}u'^\rho\mathcal{L}_{\beta'}\tau_\nu\right). \tag{4.66}$$

Dropping the prime and using the perfect fluid equations of motion in the form (2.36), we can replace the $\mathcal{L}_\beta\tau_\mu$ by $\mathcal{L}_\beta h_{\mu\nu} - h_{\nu\rho}u^\rho\mathcal{L}_\beta\tau_\mu - h_{\mu\rho}u^\rho\mathcal{L}_\beta\tau_\nu$ terms, so that we obtain (4.64) with

$$T^{\mu\nu}_{(1)\text{HS}} = \mathcal{T}^{\mu\nu}_{(1)} + \mathcal{T}^\rho_{(1)\sigma}u^\sigma X_\rho{}^{\mu\nu}, \tag{4.67}$$

where $X_\rho{}^{\mu\nu}$ is given in (2.37) and where $T^{\mu\nu}_{(1)\text{HS}}$ is in Landau frame as can be seen by comparing to (4.39).

In Appendix B we start with a constitutive relation for the non-canonical entropy current as well as the energy-momentum tensor on flat space and we work entirely on shell. We then show that without making any assumptions about the nature of the non-canonical entropy current and the energy-momentum tensor other than their constitutive relations that equation (3.11) forces the non-canonical entropy current to be of the same form as derived in this subsection. In Appendix B the functions $F_1$ and $F_2$ are replaced by $F$ and $G$ which are defined as

$$F_1 = T\left(\frac{\partial G}{\partial T}\right)_{u^2}, \tag{4.68}$$

$$F_2 = T\left(\frac{\partial G}{\partial u^2}\right)_T + \frac{T}{u^2}F. \tag{4.69}$$

This allows us to write (4.61) as

$$S^\mu_{(1)\text{non}} = (-v^\mu u^\rho + u^\mu v^\rho)\left(\partial_\rho G + \frac{1}{u^2}F\partial_\rho u^2\right). \tag{4.70}$$

Using that

$$e^{-1}\partial_\rho\left[e\left(-v^\mu u^\rho + u^\mu v^\rho\right)G\right] = Ge^{-1}\partial_\rho\left[e\left(-v^\mu u^\rho + u^\mu v^\rho\right)\right] + \left(-v^\mu u^\rho + u^\mu v^\rho\right)\partial_\rho G, \tag{4.71}$$

is identically conserved, we find

$$S^\mu_{(1)\text{non}} = Ge^{-1}\partial_\rho\left[e\left(v^\mu u^\rho - u^\mu v^\rho\right)\right] - \frac{1}{u^2}F\left(v^\mu u^\rho - u^\mu v^\rho\right)\partial_\rho u^2. \tag{4.72}$$

The flat space version of this is precisely equation (B.17).

## 4.4 Action for non-hydrostatic non-dissipative transport

In Section 4.2 we dropped the condition that $\beta^\mu$ is a Killing vector and used the hydrostatic partition function to find an action for hydrostatic non-dissipative transport. Once we drop the condition that $\beta^\mu$ is Killing we can add more terms to the action at first order in derivatives because we can no longer use the Killing equations to relate various derivatives. These extra terms can be obtained by looking at all scalars one can construct from the Lie derivatives of $\tau_\mu$ and $h_{\mu\nu}$. These are listed in equations (4.10)–(4.14). We can multiply each of these scalars by an arbitrary function forming new scalar terms that can be added to the action $S_{\text{HS}}$. Using the freedom to perform partial integrations we can drop the last term of the form $\tilde{F}e^{-1}\partial_\mu(eu^\mu)$. This leads to 4 additional terms each multiplied by one of the functions $F_3$ to $F_6$. The full first order action becomes

$$\begin{aligned}S_{(1)} = \int \mathrm{d}^{d+1}x \; e\Big(&F_1 v^\mu\partial_\mu T + F_2 v^\mu\partial_\mu u^2 + F_3 u^\mu\partial_\mu T + F_4 u^\mu\partial_\mu u^2 + F_5 u^\mu\mathcal{L}_v\tau_\mu\\&+F_6 u^\mu u^\nu\mathcal{L}_v h_{\mu\nu}\Big).\end{aligned} \tag{4.73}$$

The additional contributions to the energy current and stress tensor (in Lagrangian frame) due to the novel $F_3$ to $F_6$ contributions to the action are

$$\mathcal{T}^\mu_{F_1} = \left(\frac{\partial F_1}{\partial u^2}\right)_T u^\mu(2u^2v^\lambda\partial_\lambda T - Tv^\lambda\partial_\lambda u^2) + TF_1 u^\mu K, \tag{4.74}$$

$$\mathcal{T}^\mu_{F_2} = -\left(\frac{\partial F_2}{\partial T}\right)_{u^2} u^\mu(2u^2v^\lambda\partial_\lambda T - Tv^\lambda\partial_\lambda u^2) + 2u^2F_2 u^\mu K, \tag{4.75}$$

$$\mathcal{T}_{F_3}^{\mu} = F_3(u^{\mu} + v^{\mu})u^{\rho}\partial_{\rho}T - TF_3 u^{\mu}e^{-1}\partial_{\rho}(eu^{\rho})$$
$$- \left(\frac{\partial F_3}{\partial u^2}\right)_T u^{\mu}(Tu^{\rho}\partial_{\rho}u^2 - 2u^2 u^{\rho}\partial_{\rho}T) , \tag{4.76}$$

$$\mathcal{T}_{F_4}^{\mu} = F_4(u^{\mu} + v^{\mu})u^{\rho}\partial_{\rho}u^2 - 2u^2 F_4 u^{\mu}e^{-1}\partial_{\rho}(eu^{\rho})$$
$$+ \left(\frac{\partial F_4}{\partial T}\right)_{u^2} u^{\mu}(Tu^{\rho}\partial_{\rho}u^2 - 2u^2 u^{\rho}\partial_{\rho}T) , \tag{4.77}$$

$$\mathcal{T}_{F_5}^{\mu} = u^{\mu}u^{\sigma}v^{\rho}\tau_{\rho\sigma}\left[T\left(\frac{\partial F_5}{\partial T}\right)_{u^2} + 2u^2\left(\frac{\partial F_5}{\partial u^2}\right)_T + F_5\right] - F_5 K u^{\mu}$$
$$- F_5 e^{-1}\partial_{\rho}(eu^{\rho})v^{\mu} - F_5\mathcal{L}_u v^{\mu} + u^{\mu}v^{\rho}\partial_{\rho}F_5 - v^{\mu}u^{\rho}\partial_{\rho}F_5 , \tag{4.78}$$

$$\mathcal{T}_{F_6}^{\mu} = -2u^{\mu}u^{\rho}u^{\sigma}K_{\rho\sigma}\left[T\left(\frac{\partial F_6}{\partial T}\right)_{u^2} + 2u^2\left(\frac{\partial F_6}{\partial u^2}\right)_T + 2F_6\right] , \tag{4.79}$$

$$\mathcal{T}_{F_1}^{\mu\nu} = F_1\left(h^{\mu\nu}v^{\lambda} - h^{\lambda\mu}v^{\nu} - h^{\lambda\nu}v^{\mu}\right)\partial_{\lambda}T + 2\left(\frac{\partial F_1}{\partial u^2}\right)_T v^{\lambda}\partial_{\lambda}Tu^{\mu}u^{\nu} , \tag{4.80}$$

$$\mathcal{T}_{F_2}^{\mu\nu} = F_2\left(h^{\mu\nu}v^{\lambda} - h^{\lambda\mu}v^{\nu} - h^{\lambda\nu}v^{\mu}\right)\partial_{\lambda}u^2 + 2F_2 K u^{\mu}u^{\nu} - 2\left(\frac{\partial F_2}{\partial T}\right)_{u^2} v^{\lambda}\partial_{\lambda}Tu^{\mu}u^{\nu} , \tag{4.81}$$

$$\mathcal{T}_{F_3}^{\mu\nu} = \left(F_3 h^{\mu\nu} + 2\left(\frac{\partial F_3}{\partial u^2}\right)_T u^{\mu}u^{\nu}\right)u^{\rho}\partial_{\rho}T , \tag{4.82}$$

$$\mathcal{T}_{F_4}^{\mu\nu} = F_4 h^{\mu\nu}u^{\rho}\partial_{\rho}u^2 - 2\left(\frac{\partial F_4}{\partial T}\right)_{u^2} u^{\mu}u^{\nu}u^{\rho}\partial_{\rho}T - 2F_4 u^{\mu}u^{\nu}e^{-1}\partial_{\rho}(eu^{\rho}) , \tag{4.83}$$

$$\mathcal{T}_{F_5}^{\mu\nu} = F_5 u^{\sigma}\left(h^{\mu\nu}v^{\rho} - h^{\rho\mu}v^{\nu} - h^{\rho\nu}v^{\mu}\right)\tau_{\rho\sigma} + 2\left(\frac{\partial F_5}{\partial u^2}\right)_T u^{\mu}u^{\nu}u^{\sigma}v^{\rho}\tau_{\rho\sigma} , \tag{4.84}$$

$$\mathcal{T}_{F_6}^{\mu\nu} = -2F_6 h^{\mu\nu}u^{\rho}u^{\sigma}K_{\rho\sigma} - 4\left(\frac{\partial F_6}{\partial u^2}\right)_T u^{\mu}u^{\nu}u^{\rho}u^{\sigma}K_{\sigma\rho} - 2F_6 h^{\lambda(\mu}v^{\nu)}\left(\partial_{\lambda}u^2 - 2u^{\sigma}\mathcal{L}_u h_{\sigma\lambda}\right)$$
$$+ 4F_6 u^{(\mu}\mathcal{L}_u v^{\nu)} + 2\left(2u^{\lambda}u^{(\mu}v^{\nu)} - u^{\mu}u^{\nu}v^{\lambda}\right)\left[\left(\frac{\partial F_6}{\partial T}\right)_{u^2}\partial_{\lambda}T + \left(\frac{\partial F_6}{\partial u^2}\right)_T\partial_{\lambda}u^2\right]$$
$$+ 4F_6 u^{(\mu}v^{\nu)}e^{-1}\partial_{\lambda}(eu^{\lambda}) + 2F_6 u^{\mu}u^{\nu}K , \tag{4.85}$$

where for completeness we have included the $F_1$ and $F_2$ parts as well. These were already derived earlier in equations (4.34) and (4.35). In writing the $F_6$ part of $\mathcal{T}^{\mu\nu}$ we used the freedom to remove a term proportional to $v^{\mu}v^{\nu}$.

We will next rewrite these expressions by writing them in terms of $\mathcal{L}_{\beta}h_{\mu\nu}$ and $\mathcal{L}_{\beta}\tau_{\mu}$. Using equations (4.40), (4.41), (4.43) and

$$e^{-1}\partial_{\mu}(eu^{\mu}) = \frac{T}{2}h^{\rho\sigma}\left(\mathcal{L}_{\beta}h_{\rho\sigma} - 2u_{\rho}\mathcal{L}_{\beta}\tau_{\sigma}\right) , \tag{4.86}$$

$$\mathcal{L}_u v^{\mu} = u^{\mu}u^{\rho}v^{\nu}\tau_{\nu\rho} - Th^{\mu\nu}v^{\rho}\left(\mathcal{L}_{\beta}h_{\nu\rho} - u_{\nu}\mathcal{L}_{\beta}\tau_{\rho}\right) , \tag{4.87}$$

where we remind the reader that $u_{\mu} = h_{\mu\nu}u^{\nu}$ and $\omega_{\mu\nu} = \partial_{\mu}u_{\nu} - \partial_{\nu}u_{\mu}$. In general, $\mathcal{T}^{\mu}$ and $\mathcal{T}^{\mu\nu}$ take the following form

$$\mathcal{T}^{\mu} = \chi^{\mu\nu}\mathcal{L}_{\beta}\tau_{\nu} + \frac{1}{2}\Sigma^{\mu\nu\rho}\mathcal{L}_{\beta}h_{\nu\rho} + \frac{1}{2}\Sigma_{\text{ext}}^{\mu\nu\rho}K_{\nu\rho} , \tag{4.88}$$

$$\mathcal{T}^{\mu\nu} = \Delta^{\mu\nu\rho}\mathcal{L}_{\beta}\tau_{\rho} + \frac{1}{2}\tilde{\eta}^{\mu\nu\rho\sigma}\mathcal{L}_{\beta}h_{\rho\sigma} + \frac{1}{2}\tilde{\eta}_{\text{rot}}^{\mu\nu\rho\sigma}\omega_{\rho\sigma} + \frac{1}{2}\tilde{\eta}_{\text{ext}}^{\mu\nu\rho\sigma}K_{\rho\sigma} + \frac{1}{2}\tilde{\eta}_{\text{tor}}^{\mu\nu\rho\sigma}\tau_{\rho\sigma} , \tag{4.89}$$

where $\Sigma^{\mu\nu\rho} = \Sigma^{\mu\rho\nu}$, $\Delta^{\mu\nu\rho} = \Delta^{\nu\mu\rho}$, $\tilde{\eta}^{\mu\nu\rho\sigma} = \tilde{\eta}^{\nu\mu\rho\sigma} = \tilde{\eta}^{\mu\nu\sigma\rho}$ and similarly for the other tensors. We find that

$$\Sigma_{\text{ext}}^{\mu\nu\rho} = 2\left(TF_1 - F_5 + 2u^2 F_2\right)u^{\mu}h^{\nu\rho}$$
$$- 4\left[T\frac{\partial}{\partial T}(F_2 + F_6) - \frac{\partial}{\partial u^2}\left(TF_1 - F_5 - 2u^2 F_6\right)\right]u^{\mu}u^{\nu}u^{\rho} , \tag{4.90}$$

$$\tilde{\eta}_{\text{rot}}^{\mu\nu\rho\sigma} = 4(F_2 + F_6)v^{(\mu}h^{\nu)[\sigma}h^{\rho]\lambda}h_{\lambda\kappa}u^\kappa\,, \tag{4.91}$$

$$\tilde{\eta}_{\text{ext}}^{\mu\nu\rho\sigma} = 4(F_2 + F_6)\left[2v^{(\mu}h^{\nu)(\rho}u^{\sigma)} + u^\mu u^\nu h^{\rho\sigma} - h^{\mu\nu}u^\rho u^\sigma\right]\,, \tag{4.92}$$

$$\tilde{\eta}_{\text{tor}}^{\mu\nu\rho\sigma} = 4\left[T\frac{\partial}{\partial T}(F_2 + F_6) - \frac{\partial}{\partial u^2}(TF_1 - F_5 - 2u^2 F_6)\right]u^\mu u^\nu v^{[\rho}u^{\sigma]} + 4(F_2 + F_6)u^{(\mu}v^{\nu)}v^{[\rho}u^{\sigma]}$$
$$-2(TF_1 - F_5 + 2u^2 F_2)h^{\mu\nu}v^{[\rho}u^{\sigma]} + 4(TF_1 - F_5 + u^2(F_2 - F_6))v^{(\mu}h^{\nu)[\rho}u^{\sigma]}\,, \tag{4.93}$$

where as usual we dropped $v^\mu v^\nu$ terms and where we defined $\tilde{\eta}_{\text{rot}}^{\mu\nu\rho\sigma} = -\tilde{\eta}_{\text{rot}}^{\mu\nu\sigma\rho}$ such that $\tau_\rho \tilde{\eta}_{\text{rot}}^{\mu\nu\rho\sigma} = 0$ in order that (4.43) does not lead to any ambiguities among the various tensors. Furthermore we find that

$$\chi^{\mu\nu} = 2T\left(F_5 + T\left(\frac{\partial F_5}{\partial T}\right)_{u^2} + 2u^2\left(\frac{\partial F_5}{\partial u^2}\right)_T - 2u^2 F_4 - TF_3\right)v^{[\mu}u^{\nu]}\,, \tag{4.94}$$

$$\tilde{\eta}^{\mu\nu\rho\sigma} - \tilde{\eta}^{\rho\sigma\mu\nu} = 4T(F_2 - F_6)\left(h^{\mu\nu}u^{(\rho}v^{\sigma)} - h^{\rho\sigma}u^{(\mu}v^{\nu)}\right)$$
$$+4T(3F_6 - F_2)\left(v^{(\mu}h^{\nu)(\rho}u^{\sigma)} - v^{(\rho}h^{\sigma)(\mu}u^{\nu)}\right)$$
$$+4TF_4\left(h^{\mu\nu}u^\rho u^\sigma - h^{\rho\sigma}u^\mu u^\nu\right)$$
$$+16T\left(\frac{\partial F_6}{\partial u^2}\right)_T\left(u^\rho u^\sigma u^{(\mu}v^{\nu)} - u^\mu u^\nu u^{(\rho}v^{\sigma)}\right)\,, \tag{4.95}$$

$$\tilde{\eta}^{\mu\nu\rho\sigma} + \tilde{\eta}^{\rho\sigma\mu\nu} = 4T(F_2 + F_6)\left[-2v^{(\mu}h^{\nu)(\rho}v^{\sigma)} + \left(h^{\mu\nu}v^{(\rho}u^{\sigma)} + h^{\rho\sigma}u^{(\mu}v^{\nu)}\right)\right.$$
$$\left.-\left(v^{(\mu}h^{\nu)(\rho}u^{\sigma)} + v^{(\rho}h^{\sigma)(\mu}u^{\nu)}\right)\right]\,, \tag{4.96}$$

$$\Delta^{\mu\nu\rho} - \Sigma^{\rho\mu\nu} = 2T\left(TF_1 - F_5 + u^2(F_2 - F_6)\right)h^{\rho(\mu}v^{\nu)} - T\left(TF_1 - F_5 + 2u^2 F_2\right)v^\rho h^{\mu\nu}$$
$$+2T(F_2 + F_6)(v^\rho + u^\rho)u^{(\mu}v^{\nu)} + 2T^2\frac{\partial}{\partial T}(F_2 + F_6)\left(v^\rho u^\mu u^\nu - 2u^\rho u^{(\mu}v^{\nu)}\right)$$
$$-2T\frac{\partial}{\partial u^2}(TF_1 - F_5 - 2u^2 F_6)\left(v^\rho u^\mu u^\nu - 2u^\rho u^{(\mu}v^{\nu)}\right)\,, \tag{4.97}$$

$$\Delta^{\mu\nu\rho} + \Sigma^{\rho\mu\nu} = 2T\left(TF_1 + F_5 + 2u^2 F_2 - u^2(F_2 + F_6)\right)h^{\rho(\mu}v^{\nu)} - T\left(TF_1 + F_5 + 2u^2 F_2\right)v^\rho h^{\mu\nu}$$
$$+2T(F_2 + F_6)v^\rho u^{(\mu}v^{\nu)} - 2T(TF_3 + 2u^2 F_4)u^\rho h^{\mu\nu}$$
$$+4T\left(F_4 + T\left(\frac{\partial F_4}{\partial T}\right)_{u^2} - T\left(\frac{\partial F_3}{\partial u^2}\right)_T\right)u^\rho u^\mu u^\nu$$
$$+2T\left(-\frac{\partial}{\partial u^2}(TF_1 + F_5 - 2u^2 F_6) + T\frac{\partial}{\partial T}(F_2 + F_6) + 2F_4\right)v^\rho u^\mu u^\nu$$
$$+2T\left(-2\frac{\partial}{\partial u^2}(TF_1 - F_5 + 2u^2 F_6) + 2T\frac{\partial}{\partial T}(F_2 - F_6) + F_2 + F_6\right)u^\rho u^{(\mu}v^{\nu)}\,, \tag{4.98}$$

where we have discarded terms in $\mathcal{T}^{\mu\nu}$ proportional to $v^\mu v^\nu$.

As we have seen in Section 3 the NHS terms are defined as those contributions to the energy-momentum tensor for which

$$-\mathcal{T}^\mu\mathcal{L}_\beta\tau_\mu + \frac{1}{2}\mathcal{T}^{\mu\nu}\mathcal{L}_\beta h_{\mu\nu} = 0\,. \tag{4.99}$$

Using equations (4.88) and (4.89) we can see that for this to be the case it is necessary that $\Sigma_{\text{ext}}^{\mu\nu\rho}$, $\tilde{\eta}_{\text{rot}}^{\mu\nu\rho\sigma}$, $\tilde{\eta}_{\text{ext}}^{\mu\nu\rho\sigma}$ and $\tilde{\eta}_{\text{tor}}^{\mu\nu\rho\sigma}$ all vanish. The $\chi^{\mu\nu}$ is anti-symmetric so there are no $\mathcal{L}_\beta\tau_\mu$ squared contributions to (4.99). The condition $\tilde{\eta}^{\mu\nu\rho\sigma} + \tilde{\eta}^{\rho\sigma\mu\nu} = 0$ guarantees that the $\tilde{\eta}^{\mu\nu\rho\sigma}$ is anti-symmetric under the interchange of the first pair of symmetric indices with the second pair of symmetric indices ensuring that there are no $\mathcal{L}_\beta h_{\mu\nu}$ squared contributions to (4.99). Finally, cancellation of the cross terms $\mathcal{L}_\beta h_{\mu\nu}\mathcal{L}_\beta\tau_\rho$ requires that we set $\Delta^{\mu\nu\rho} - \Sigma^{\rho\mu\nu} = 0$. As one can see by inspection all of these conditions will be obeyed provided we set

$$F_1 = \frac{1}{T}\left(F_5 + 2u^2 F_6\right)\,, \qquad F_2 = -F_6\,. \tag{4.100}$$

We thus conclude that when (4.100) holds, equation (4.99) holds off shell. Setting $F_1$ and $F_2$ equal to their NHS values we obtain the following action for pure NHS transport at first order

$$S_{\text{NHS}} = \int d^{d+1}x\, e\left(F_3 u^\mu \partial_\mu T + F_4 u^\mu \partial_\mu u^2 - T F_5 v^\mu \mathcal{L}_\beta \tau_\mu - 2 T F_6 u^\mu v^\nu \mathcal{L}_\beta h_{\mu\nu}\right). \tag{4.101}$$

The NHS currents are then schematically

$$\mathcal{T}^\mu_{\text{NHS}} = \sum_{i=1}^6 \mathcal{T}^\mu_{F_i}|_{F_1 = T^{-1}F_5 + 2T^{-1}u^2 F_6;\; F_2 = -F_6}, \tag{4.102}$$

$$\mathcal{T}^{\mu\nu}_{\text{NHS}} = \sum_{i=1}^6 \mathcal{T}^{\mu\nu}_{F_i}|_{F_1 = T^{-1}F_5 + 2T^{-1}u^2 F_6;\; F_2 = -F_6}. \tag{4.103}$$

On shell and in Landau frame we have for $\mathcal{T}^\mu = \sum_{i=1}^6 \mathcal{T}^\mu_{F_i}$ and $\mathcal{T}^{\mu\nu} = \sum_{i=1}^6 \mathcal{T}^{\mu\nu}_{F_i}$ that

$$\begin{aligned}
T^{\mu\nu}_{\text{HS}} + T^{\mu\nu}_{\text{NHS}} &= \mathcal{T}^{\mu\nu} + \mathcal{T}^{\rho\sigma} u_\sigma X_\rho{}^{\mu\nu} - \mathcal{T}^\rho X_\rho{}^{\mu\nu} = \frac{1}{2}\eta^{\mu\nu\alpha\beta}\left(\mathcal{L}_\beta h_{\alpha\beta} - u_\alpha \mathcal{L}_\beta \tau_\beta - u_\beta \mathcal{L}_\beta \tau_\alpha\right) \\
&\quad + \frac{1}{2}\eta^{\mu\nu\alpha\beta}_{\text{rot}} \omega_{\alpha\beta} + \frac{1}{2}\eta^{\mu\nu\alpha\beta}_{\text{ext}} K_{\alpha\beta} + \frac{1}{2}\eta^{\mu\nu\alpha\beta}_{\text{tor}} \tau_{\alpha\beta},
\end{aligned} \tag{4.104}$$

where we remind the reader that $T^{\mu\nu}$ is defined in equation (4.33). In here the tensors are given by

$$\begin{aligned}
\eta^{\mu\nu\alpha\beta} &= \tilde{\eta}^{\mu\nu\alpha\beta} + \left(-\Sigma^{\rho\alpha\beta} + \tilde{\eta}^{\rho\sigma\alpha\beta} u_\sigma\right)X_\rho{}^{\mu\nu} + (\Delta^{\mu\nu\rho} + \tilde{\eta}^{\mu\nu\rho\sigma} u_\sigma)X_\rho{}^{\alpha\beta} \\
&\quad + \left[-\chi^{\rho\sigma} + \left(\Delta^{\rho\lambda\sigma} - \Sigma^{\rho\lambda\sigma}\right)u_\lambda + \tilde{\eta}^{\rho\kappa\sigma\lambda} u_\kappa u_\lambda\right]X_\rho{}^{\mu\nu} X_\sigma{}^{\alpha\beta}, \tag{4.105} \\
\eta^{\mu\nu\alpha\beta}_{\text{rot}} &= \tilde{\eta}^{\mu\nu\alpha\beta}_{\text{rot}} + \tilde{\eta}^{\rho\sigma\alpha\beta}_{\text{rot}} u_\sigma X_\rho{}^{\mu\nu}, \tag{4.106} \\
\eta^{\mu\nu\alpha\beta}_{\text{ext}} &= \tilde{\eta}^{\mu\nu\alpha\beta}_{\text{ext}} + \tilde{\eta}^{\rho\sigma\alpha\beta}_{\text{ext}} u_\sigma X_\rho{}^{\mu\nu} - \Sigma^{\rho\alpha\beta}_{\text{ext}} X_\rho{}^{\mu\nu}, \tag{4.107} \\
\eta^{\mu\nu\alpha\beta}_{\text{tor}} &= \tilde{\eta}^{\mu\nu\alpha\beta}_{\text{tor}} + \tilde{\eta}^{\rho\sigma\alpha\beta}_{\text{tor}} u_\sigma X_\rho{}^{\mu\nu}. \tag{4.108}
\end{aligned}$$

The pure NHS part in Landau frame is given by

$$T^{\mu\nu}_{\text{NHS}} = \frac{1}{2}\eta^{\mu\nu\alpha\beta}_{\text{NHS}}\left(\mathcal{L}_\beta h_{\alpha\beta} - u_\alpha \mathcal{L}_\beta \tau_\beta - u_\beta \mathcal{L}_\beta \tau_\alpha\right), \tag{4.109}$$

where $\eta^{\mu\nu\alpha\beta}_{\text{NHS}} = -\eta^{\alpha\beta\mu\nu}_{\text{NHS}}$ is obtained by substituting (4.100) into (4.105).

One might wonder what the expression for the pure HS part is. However, for the HS sector it is only the symmetric part of $\eta^{\mu\nu\alpha\beta}$ as well as the objects $\eta^{\mu\nu\rho\sigma}_{\text{rot}}$, $\eta^{\mu\nu\rho\sigma}_{\text{ext}}$ and $\eta^{\mu\nu\rho\sigma}_{\text{tor}}$ that are uniquely determined. These all depend on two functions $F_2 + F_6$ and $T F_1 - F_5 + 2u^2 F_2$. There is no unique HS expression for the remaining four functions in the action. The reason behind this is that we know that

$$-\mathcal{T}^\mu_{\text{HS}} \mathcal{L}_\beta \tau_\mu + \frac{1}{2}\mathcal{T}^{\mu\nu}_{\text{HS}} \mathcal{L}_\beta h_{\mu\nu} = e^{-1}\partial_\mu\left(e S^\mu_{\text{non}}\right), \tag{4.110}$$

but this only uniquely fixes the symmetric part of $\eta^{\mu\nu\rho\sigma}$ as well as the extrinsic, torsion and rotation $\eta$-tensors. The anti-symmetric part cannot be fixed. This freedom is precisely encoded by the NHS terms. In a sense the HS coefficients belong to the 'quotient space' of non-dissipative transport coefficients modulo the NHS ones. Hence two HS transport coefficients are equivalent if they differ by an NHS term. In Section 4.2 we picked a representative of the HS sector by setting $F_3 = F_4 = F_5 = F_6 = 0$.

The second line in (4.105) is anti-symmetric under interchanging the pair $\mu\nu$ with $\alpha\beta$ as can be seen from the fact that

$$-\chi^{\rho\sigma} + \left(\Delta^{\rho\lambda\sigma} - \Sigma^{\rho\lambda\sigma}\right)u_\lambda + \tilde{\eta}^{\rho\kappa\sigma\lambda}u_\kappa u_\lambda =$$
$$2T\left[2Tu^2\left(\frac{\partial F_6}{\partial T}\right)_{u^2} - 2TF_1 - 2u^2 F_6 + 2F_5 + 4u^2\left(\frac{\partial F_5}{\partial u^2}\right)_T + T\left(\frac{\partial F_5}{\partial T}\right)_{u^2}\right]u^{[\rho}v^{\sigma]}.$$

(4.111)

This means that the symmetric part of $\eta^{\mu\nu\alpha\beta}$ does not contain terms that are quadratic in $X_\rho{}^{\mu\nu}$. This explains why the $\mathcal{J}$ coefficients in (4.44) do not contain product of $\rho$ and $P$ and/or derivatives thereof, while the $\mathcal{A}$ coefficients do admit such terms.

# 5 First order corrections

This section can be viewed as a continuation of Section 3, in which we use constitutive relations and non-negativity of entropy production to find all the allowed first order corrections to the boost-agnostic perfect fluid energy-momentum tensor. We already dealt with the constitutive relations for the HS sector in Appendix B and so we will only be concerned with the constitutive relations for NHS and dissipative transport. We also show how to recover Lifshitz fluids as well as Lorentz boost invariant fluids from our general framework. Finally, we consider the limit of small fluid velocity and show how our formalism recovers the results of [11].

## 5.1 Constitutive relations

Using our result (4.64), the relation (A.11) tells us that to second order in derivatives and in Landau frame,

$$e^{-1}\partial_\mu(eS^\mu) = -\frac{1}{2T}\left(T_{(1)}^{\mu\nu} - T_{(1)\text{HS}}^{\mu\nu}\right)\left(\mathcal{L}_u h_{\mu\nu} - h_{\rho\nu}u^\rho\mathcal{L}_u\tau_\mu - h_{\mu\rho}u^\rho\mathcal{L}_u\tau_\nu\right), \qquad (5.1)$$

where $T_{(1)\text{HS}}^{\mu\nu}$ is the Landau frame hydrostatic contribution as defined in (3.8), and $T_{(1)}^{\mu\nu}$ is the full energy-momentum tensor in Landau frame.

Since the divergence of the entropy current is a quadratic form in the derivatives of the fluid variables, equation (5.1) tells us which derivatives we should use to write the constitutive relations for the energy-momentum tensor. The fluid variables are $\mathcal{L}_u\tau_\mu$ and $\mathcal{L}_u h_{\mu\nu}$, and we may thus write the following constitutive relation for the part of the energy-momentum tensor that is not of hydrostatic origin[12]

$$T_{(1)}^{\mu\nu} - T_{(1)\text{HS}}^{\mu\nu} = \frac{1}{2}\eta^{\mu\nu\rho\sigma}\mathcal{L}_u h_{\rho\sigma} + \tilde{\zeta}^{\mu\nu\rho}\mathcal{L}_u\tau_\rho. \qquad (5.2)$$

By redefining $\tilde{\zeta}^{\mu\nu\rho}$, this can be written equivalently as

$$T_{(1)}^{\mu\nu} - T_{(1)\text{HS}}^{\mu\nu} = \frac{1}{2}\eta^{\mu\nu\rho\sigma}\left(\mathcal{L}_u h_{\rho\sigma} - h_{\kappa\sigma}u^\kappa\mathcal{L}_u\tau_\rho - h_{\rho\kappa}u^\kappa\mathcal{L}_u\tau_\sigma\right) + \zeta^{\mu\nu\rho}\mathcal{L}_u\tau_\rho. \qquad (5.3)$$

Upon substituting the constitutive relations into the right hand side of (5.1) we obtain a quadratic form. Non-negative entropy production will restrict the form of the $\eta^{\mu\nu\rho\sigma}$ tensor, and it tells us that $\zeta^{\mu\nu\rho}$ must vanish. This is because there are no $\mathcal{L}_u\tau_\rho$ squared terms in (5.1). Furthermore, terms involving the anti-symmetric combination of velocity derivatives

---

[12]The $\eta$-tensor in (5.2) should not be confused with the $\eta$-tensor that appears in the context of Lagrangian transport in (4.104).

(analogous to the $\eta^{\mu\nu\rho\sigma}_{\text{rot}}$ term in (4.42)) are also explicitly forbidden by the requirement that the divergence of the entropy current is a quadratic form. We thus conclude that

$$T^{\mu\nu}_{(1)} - T^{\mu\nu}_{(1)\text{HS}} = \frac{1}{2}\eta^{\mu\nu\rho\sigma}\left(\mathcal{L}_u h_{\rho\sigma} - h_{\kappa\sigma}u^\kappa \mathcal{L}_u \tau_\rho - h_{\rho\kappa}u^\kappa \mathcal{L}_u \tau_\sigma\right), \tag{5.4}$$

so that all that is left is to classify all the allowed terms that make up $\eta^{\mu\nu\rho\sigma}$. This can be achieved by looking at the symmetries of the fluid.

In addition to the $SO(d)$ Ward identity (which is manifest in the symmetry of $T^{\mu\nu}$) the energy-momentum tensor must respect the symmetries of the thermal state around which we expand. In the absence of boost symmetries the thermal state spontaneously breaks the $SO(d)$ symmetry down to the $SO(d-1)$ subgroup that preserves the velocity $h^{\mu\rho}h_{\rho\nu}u^\nu$. In flat space these are the rotations preserving $v^i$. In other words, different absolute values of velocities correspond to different thermodynamic states of the theory.

Therefore, the natural tensor structures are the $SO(d-1)$ invariant tensors $v^\mu$ as well as

$$P^{\mu\nu} = h^{\mu\nu} - n^\mu n^\nu, \qquad n^\mu = \frac{h^{\mu\nu}h_{\nu\rho}u^\rho}{\sqrt{u^2}}, \tag{5.5}$$

where $n^\mu n^\nu h_{\mu\nu} = 1$. The tensor $P^{\mu\nu}$ is a projector onto the space orthogonal to the unit vector $n^\mu$. In terms of these tensor structures, the constitutive relation takes the form[13,14]

$$\begin{aligned}
\eta^{\mu\nu\rho\sigma} =& \mathfrak{t}\left(P^{\mu\rho}P^{\nu\sigma} + P^{\mu\sigma}P^{\nu\rho} - \frac{2}{d-1}P^{\mu\nu}P^{\rho\sigma}\right) \\
& + \frac{4s_1}{u^2}v^{(\mu}n^{\nu)}n^{(\rho}v^{\sigma)} + s_2 n^\mu n^\nu n^\rho n^\sigma + s_3 P^{\mu\nu}P^{\rho\sigma} \\
& + \frac{4f_1}{u^2}v^{(\mu}P^{\nu)(\rho}v^{\sigma)} + f_2\left(P^{\mu\rho}n^\nu n^\sigma + P^{\nu\rho}n^\mu n^\sigma + P^{\mu\sigma}n^\nu n^\rho + P^{\nu\sigma}n^\mu n^\rho\right) \\
& + s_6\left(P^{\rho\sigma}n^\mu n^\nu + P^{\mu\nu}n^\rho n^\sigma\right) - s_3^{\text{NHS}}\left(P^{\rho\sigma}n^\mu n^\nu - P^{\mu\nu}n^\rho n^\sigma\right) \\
& - \frac{4f_3}{\sqrt{u^2}}\left(v^{(\mu}P^{\nu)(\rho}n^{\sigma)} + n^{(\mu}P^{\nu)(\rho}v^{\sigma)}\right) - \frac{4f^{\text{NHS}}}{\sqrt{u^2}}\left(v^{(\mu}P^{\nu)(\rho}n^{\sigma)} - n^{(\mu}P^{\nu)(\rho}v^{\sigma)}\right) \\
& - \frac{2s_5}{\sqrt{u^2}}\left(v^{(\mu}n^{\nu)}P^{\rho\sigma} + P^{\mu\nu}n^{(\rho}v^{\sigma)}\right) - \frac{2s_1^{\text{NHS}}}{\sqrt{u^2}}\left(v^{(\mu}n^{\nu)}P^{\rho\sigma} - P^{\mu\nu}n^{(\rho}v^{\sigma)}\right) \\
& - \frac{2s_4}{\sqrt{u^2}}\left(v^{(\mu}n^{\nu)}n^\rho n^\sigma + n^\mu n^\nu n^{(\rho}v^{\sigma)}\right) - \frac{2s_2^{\text{NHS}}}{\sqrt{u^2}}\left(v^{(\mu}n^{\nu)}n^\rho n^\sigma - n^\mu n^\nu n^{(\rho}v^{\sigma)}\right), \tag{5.6}
\end{aligned}$$

leading to a total of 14 transport coefficients. The $f_1$ term in was also observed in [13]. We see that the $\eta$-tensor has a part that is anti-symmetric under interchanging the pairs of symmetric indices. This is related to non-hydrostatic non-dissipative transport and is the topic of Section 5.2. This leaves 10 coefficients that could contribute to dissipative transport. The normalization of the 14 coefficients has been chosen such that all coefficients have the same scaling dimension which is $d$, the number of spatial dimensions. We note that $h^{\mu\nu}$ will be assigned a scaling dimension of 2 while $v^\mu$ and $u^\mu$ will have scaling dimension $z$.

A unique feature of Landau frame is that the derivative corrections to the energy current is given entirely in terms of the $(2,0)$ momentum-stress tensor (cf. the second relation in (A.6)), which in turn means that the $(1,1)$ energy-momentum tensor (2.24) at first

---

[13]We remark again that this object has two redundancies: we can add to $\eta^{\mu\nu\rho\sigma}$ any term of the form $v^\mu v^\nu Y^{\rho\sigma}$ or $v^\rho v^\sigma Z^{\mu\nu}$ for arbitrary $Y^{\mu\nu}$ and $Z^{\mu\nu}$ without changing $T^\mu{}_\nu$.

[14]The rationale behind the naming scheme we have adopted for the transport coefficients will become apparent in the next section (see in particular Eq. (5.22)), where we show that in the expression for the divergence of the entropy current, the coefficients $\{s_1, s_2 \dots\}$ multiply scalar structures, the coefficients $\{f_1, f_2, \dots\}$ multiply vector structures, while, finally, the coefficient $\mathfrak{t}$ multiplies a single tensor structure.

order can be constructed from the $(2,0)$ momentum-stress tensor. More precisely, defining $T_{(1)D,NHS}^{\mu\nu} := T_{(1)D}^{\mu\nu} + T_{(1)NHS}^{\mu\nu} = T_{(1)}^{\mu\nu} - T_{(1)HS}^{\mu\nu}$, where each term is a symmetric tensor, we have

$$(T_{(1)D,NHS})^\mu{}_\nu = \frac{1}{2}\left[-\eta^{\mu\rho\kappa\lambda}u_\rho\tau_\nu + \eta^{\mu\rho\kappa\lambda}h_{\rho\nu}\right](\mathcal{L}_u h_{\kappa\lambda} - u_\kappa\mathcal{L}_u\tau_\lambda - u_\lambda\mathcal{L}_u\tau_\kappa), \quad (5.7)$$

where we have used the relation (2.24). On flat space (2.19), where $u^\mu = (1, v^i)$, the energy-momentum tensor – and by extension the tensor $\eta^{\mu\nu\rho\sigma}$ – may be further decomposed as,

$$(T_{(1)D,NHS})^0{}_j = \frac{1}{2}\eta_{jkl}\left(\partial_k v^l + \partial_l v^k\right) + \kappa_{jk}\partial_t v^k, \quad (5.8)$$

$$(T_{(1)D,NHS})^i{}_j = \frac{1}{2}\eta^{ijkl}\left(\partial_k v^l + \partial_l v^k\right) + \kappa^{ijk}\partial_t v^k, \quad (5.9)$$

where the flat space tensors $\kappa_{jk}$, $\eta_{jkl}$, $\kappa^{ijk}$ are given by

$$\kappa_{jk} = \eta^{0j0k}, \qquad \eta_{jkl} = \eta^{0jkl}, \qquad \kappa^{ijk} = \eta^{ijk0}, \quad (5.10)$$

which means that

$$\kappa_{jk} = \frac{f_1}{v^2}P_{jk} + \frac{s_1}{v^2}n_j n_k, \quad (5.11)$$

$$\eta_{jkl} = \frac{f_3 + f^{NHS}}{\sqrt{v^2}}\left(P^{jk}n^l + P^{jl}n^k\right) + \frac{s_5 + s_1^{NHS}}{\sqrt{v^2}}P^{kl}n^j + \frac{s_4 + s_2^{NHS}}{\sqrt{v^2}}n^j n^k n^l, \quad (5.12)$$

$$\kappa^{ijk} = \frac{f_3 - f^{NHS}}{\sqrt{v^2}}\left(P^{jk}n^i + P^{ik}n^j\right) + \frac{s_5 - s_1^{NHS}}{\sqrt{v^2}}P^{ij}n^k + \frac{s_4 - s_2^{NHS}}{\sqrt{v^2}}n^i n^j n^k, \quad (5.13)$$

$$\begin{aligned}
\eta^{ijkl} = {}& \mathfrak{t}\left(P^{ik}P^{jl} + P^{il}P^{jk} - \frac{2}{d-1}P^{ij}P^{kl}\right) + s_3 P^{ij}P^{kl} \\
& + f_2\left(P^{ik}n^j n^l + P^{jk}n^i n^l + P^{il}n^j n^k + P^{jl}n^i n^k\right) + s_2 n^i n^j n^k n^l \\
& + s_6\left(P^{kl}n^i n^j + P^{ij}n^k n^l\right) + s_3^{NHS}\left(P^{ij}n^k n^l - P^{kl}n^i n^j\right),
\end{aligned} \quad (5.14)$$

where the result has been written in terms of

$$n^i = \frac{v^i}{\sqrt{v^2}}, \qquad P^{ij} = \delta^i_j - \frac{v^i v^j}{v^2} = \delta^i_j - n^i n^j, \quad (5.15)$$

which are the flat space versions of (5.5).

## 5.2 Non-hydrostatic non-dissipative transport & Onsager relations

The subsector of transport obtained by isolating the anti-symmetric part of $\eta$, i.e. $\eta_A^{\mu\nu\rho\sigma} \subset \eta^{\mu\nu\rho\sigma}$ with $\eta_A^{\mu\nu\rho\sigma} = -\eta_A^{\rho\sigma\mu\nu}$, corresponds to the non-hydrostatic (NHS) non-dissipative transport. By using (5.7) and (5.1), such terms trivially produce no entropy. The constitutive relations tell us that there are at most 4 transport coefficients of this type. In Section 4.4 we found precisely 4 terms in the action that corresponded to the NHS sector. Extracting the anti-symmetric part of (5.6), we get

$$\begin{aligned}
\eta_A^{\mu\nu\rho\sigma} = {}& \frac{4f^{NHS}}{\sqrt{u^2}}\left(n^{(\mu}P^{\nu)(\rho}v^{\sigma)} - n^{(\rho}P^{\sigma)(\mu}v^{\nu)}\right) + \frac{2s_1^{NHS}}{\sqrt{u^2}}\left(P^{\mu\nu}n^{(\rho}v^{\sigma)} - P^{\rho\sigma}n^{(\mu}v^{\nu)}\right) \\
& + \frac{2s_2^{NHS}}{\sqrt{u^2}}\left(n^\mu n^\nu n^{(\rho}v^{\sigma)} - n^\rho n^\sigma n^{(\mu}v^{\nu)}\right) + s_3^{NHS}\left(P^{\mu\nu}n^\rho n^\sigma - P^{\rho\sigma}n^\mu n^\nu\right).
\end{aligned} \quad (5.16)$$

Demanding the absence of NHS transport is, at linear order, equivalent to the Onsager relations [59,60], which express the fact that there are no anti-symmetric contributions to the

$\eta$-tensor in systems with time-reversal symmetry. More explicitly, consider linearized perturbations around global thermal equilibrium $(T_0, v_0^i)$ in flat space,

$$v^i = v_0^i + \delta v^i, \qquad T = T_0 + \delta T \,,$$

where $\delta v^i$ and $\delta T$ are the fluctuations. The resulting change in the energy-momentum tensor to first order in fluctuations is

$$\delta T_{(1)}^{\mu\nu} = \frac{1}{2}\eta_0^{\mu\nu\rho\sigma}(\partial_\rho \delta u_\sigma + \partial_\sigma \delta u_\rho)\,,$$

where $\delta u^\mu = (0, \delta v^i)$ and indices are lowered by $h_{\mu\nu} = \delta_\mu^i \delta_\nu^i$. The Onsager relations then tell us that[15]

$$\eta_0^{\mu\nu\rho\sigma} = \eta_0^{\rho\sigma\mu\nu}\,,$$

which is the linearized version of the general requirement of symmetry, $\eta^{\mu\nu\rho\sigma} = \eta^{\rho\sigma\mu\nu}$.

Imposing symmetry on the $\eta$-tensor is equivalent to the vanishing of all NHS coefficients,

$$f^{\mathrm{NHS}} = s_1^{\mathrm{NHS}} = s_2^{\mathrm{NHS}} = s_3^{\mathrm{NHS}} = 0 \,. \tag{5.17}$$

Hence, ignoring the NHS sector, we obtain the following constitutive relations for the dissipative sector

$$\kappa_{jk} = \frac{f_1}{v^2}P_{jk} + \frac{s_1}{v^2}n_j n_k\,, \tag{5.18}$$

$$\kappa^{ijk} = \frac{f_3}{\sqrt{v^2}}\left(P^{jk}n^i + P^{ik}n^j\right) + \frac{1}{\sqrt{v^2}}s_5 P^{ij}n^k + \frac{1}{\sqrt{v^2}}s_4 n^i n^j n^k\,, \tag{5.19}$$

$$\begin{aligned}
\eta^{ijkl} = {}& \mathfrak{t}\left(P^{ik}P^{jl} + P^{il}P^{jk} - \frac{2}{d-1}P^{ij}P^{kl}\right) \\
& + f_2\left(P^{ik}n^j n^l + P^{jk}n^i n^l + P^{il}n^j n^k + P^{jl}n^i n^k\right) \\
& + s_3 P^{ij}P^{kl} + s_6\left(P^{kl}n^i n^j + P^{ij}n^k n^l\right) + s_2 n^i n^j n^k n^l\,,
\end{aligned} \tag{5.20}$$

leaving us with 10 candidate coefficients for dissipative transport. Note that in the absence of NHS terms we have the identity $\eta_{jkl} = \eta^{0jkl} = \eta^{klj0} = \kappa^{klj}$ (cf. (5.10) and (5.11)–(5.14)). For the remainder of this section we will be working in flat space. In the next subsection we will show that all these coefficients contribute to dissipation provided they obey suitable inequalities.

## 5.3 Dissipative transport

In this section, we derive additional constraints on the dissipative transport coefficients from the requirement of positivity of entropy production. Using our results (5.18)–(5.20), the divergence of the entropy current in Landau frame (5.1) on flat space along with the requirement that it be positive definite for the dissipative sector reads

$$\begin{aligned}
-T\partial_\mu S^\mu = {}& \frac{1}{4}\eta^{ijkl}\left(\partial_i v^j + \partial_j v^i\right)\left(\partial_k v^l + \partial_l v^k\right) + \kappa_{ij}\partial_t v^i \partial_t v^j \\
& + \kappa^{ijk}\left(\partial_i v^j + \partial_j v^i\right)\partial_t v^k \leq 0\,,
\end{aligned} \tag{5.21}$$

with equality if and only if all the dissipative coefficients are zero.

---

[15] One way to see this is because $\langle \partial_\mu \delta u_\nu(t)\delta T_{(1)}^{\mu\nu}(0)\rangle = \langle \partial_\mu \delta u_\nu(0)\delta T_{(1)}^{\mu\nu}(t)\rangle$, which is a direct result of time reversal symmetry.

It is useful to decompose the expression (5.21) into scalar, vector and tensor sectors,

$$\vec{b}^{(S)T} A^{(S)}_{(3\times3)} \vec{b}^{(S)} + \vec{b}^{(V)T}_i A^{(V)}_{(2\times2)} \vec{b}^{(V)}_i + \mathfrak{t}\, A^{(T)} \leq 0 \,, \tag{5.22}$$

where the basis vector of scalars is

$$\vec{b}^{(S)} = \begin{pmatrix} \frac{1}{2}\frac{1}{v^2}\partial_t v^2 \\ n^i n^j \partial_i v^j \\ P^{ij}\partial_i v^j \end{pmatrix} \,, \tag{5.23}$$

with the associated quadratic form

$$A^{(S)}_{(3\times3)} = \begin{pmatrix} s_1 & s_4 & s_5 \\ s_4 & s_2 & s_6 \\ s_5 & s_6 & s_3 \end{pmatrix} \,. \tag{5.24}$$

The basis vector of vectors is

$$\vec{b}^{(V)}_i = \begin{pmatrix} \frac{1}{\sqrt{v^2}}\partial_t v^i \\ P^{ij} n^k (\partial_j v^k + \partial_k v^j) \end{pmatrix} \,, \tag{5.25}$$

which has the associated quadratic form

$$A^{(V)}_{(2\times2)} = \begin{pmatrix} f_1 & f_3 \\ f_3 & f_2 \end{pmatrix} \,. \tag{5.26}$$

The single tensor structure is described by

$$A^{(T)} = \left( P^{ik}P^{jl} + P^{il}P^{jk} - \frac{2}{d-1}P^{ij}P^{kl} \right) \left( \partial_i v^j + \partial_j v^i \right) \left( \partial_k v^l + \partial_l v^k \right) \,. \tag{5.27}$$

Positivity of entropy production must hold for all fluid configurations and thus for each of the three sectors separately.

The tensor contribution requires that

$$\mathfrak{t} \leq 0 \,. \tag{5.28}$$

The quadratic form of the vector sector must be negative definite, i.e. all its eigenvalues must be negative, which is the case if and only if

$$f_1 \leq 0 \,, \qquad f_2 \leq 0 \,, \qquad \det A^{(V)}_{(2\times2)} = f_1 f_2 - f_3^2 \geq 0 \,. \tag{5.29}$$

Finally, the quadratic form of the scalar sector must be negative definite as well, which gives the conditions

$$s_1 \leq 0 \,, \qquad s_2 \leq 0 \,, \qquad s_3 \leq 0 \,, \tag{5.30}$$

$$s_1 s_2 - s_4^2 \geq 0 \,, \qquad s_1 s_3 - s_5^2 \geq 0 \,, \qquad s_3 s_2 - s_6^2 \geq 0 \,, \tag{5.31}$$

$$\det A^{(S)}_{(3\times3)} = s_1 s_2 s_3 - s_3 s_4^2 - s_2 s_5^2 + 2 s_4 s_5 s_6 - s_1 s_6^2 \leq 0 \,. \tag{5.32}$$

## 5.4 Scale invariance: Lifshitz fluid dynamics

If our fluid enjoys scale symmetry with dynamical exponent $z$, the following Ward identity must be satisfied (see also [9] for more details)

$$z T^0{}_0 + T^i{}_i = 0. \tag{5.33}$$

Using the constitutive relations for the dissipative sector (5.18)–(5.20), this gives rise to the following relations at first order

$$z v^j \kappa_{jk} = \delta_{ij} \kappa^{ijk}, \qquad z v^j \eta_{jkl} = \delta_{ij} \eta^{ijkl}, \tag{5.34}$$

which amounts to

$$\begin{aligned} z s_5 &= (d-1) s_3 + s_6, \\ z s_4 &= (d-1) s_6 + s_2, \\ z s_1 &= (d-1) s_5 + s_4. \end{aligned} \tag{5.35}$$

Hence, if we impose scale symmetry, the 6 dissipative scalar transport coefficients get reduced by 3, while the 3 vector and 1 tensor transport coefficients are unaffected. In other words, uncharged Lifshitz hydrodynamics has 7 dissipative transport coefficients at first order. We note that the inequality type constraints for Lifshitz fluids are obtained by substituting the relations (5.35) into (5.30)–(5.32).

We furthermore remark that since all transport coefficients that appear in the $\eta$-tensor (5.6) are functions of $T$ and $v^2$ and have scaling dimension $d$, they must be of the form

$$T^{d/z} f(\alpha), \qquad \alpha = v^2 T^{\frac{2}{z}-2}, \tag{5.36}$$

for some unknown function $f$, where $\alpha$ has no scaling dimension.

Turning to the NHS sector, which is described by (5.16), we find that scale symmetry gets rid of two coefficients. It sets $s_2^{\text{NHS}} = -(d-1)s_1^{\text{NHS}}$ and $s_3^{\text{NHS}} = -z s_1^{\text{NHS}}$ and leaves $f^{\text{NHS}}$ free. Furthermore, scale symmetry reduces the number of hydrostatic transport coefficients from two to one. This can be seen as follows. The $z$-trace Ward identity on curved space reads

$$-z v^\nu \tau_\mu T^\mu{}_\nu + h_{\mu\rho} h^{\rho\nu} T^\mu{}_\nu = -z \tau_\mu T^\mu + h_{\mu\nu} T^{\mu\nu} = 0. \tag{5.37}$$

If we substitute (4.34) and (4.35) as well as (4.37) into this Ward identity we find that (by looking at the terms proportional to the trace of the extrinsic curvature)

$$F_1 = -2 F_2 \frac{u^2(z-1)}{zT}. \tag{5.38}$$

The rest of the terms in (5.37) then tell us that

$$P(T, u^2) = T^{1+\frac{d}{z}} p(\alpha), \qquad F_2(T, u^2) T^{2-\frac{2}{z}} = T^{\frac{d}{z}} q(\alpha), \tag{5.39}$$

where $p(\alpha)$ and $q(\alpha)$ are arbitrary functions of $\alpha$ which is the scale invariant combination $\alpha = u^2 T^{\frac{2}{z}-2}$. We can then write the hydrostatic partition function (4.17) as

$$S_{\text{HS(Lif)}} = \int d^{d+1}x\, e \left( T^{1+\frac{d}{z}} p(\alpha) + T^{\frac{d}{z}} q(\alpha) v^\mu \partial_\mu \alpha \right) + \mathcal{O}(\partial^2). \tag{5.40}$$

The same can be done for the action (4.101) describing NHS transport, which for Lifshitz scaling takes the form

$$S_{\text{NHS(Lif)}} = \int d^{d+1}x\, e \left( T^{\frac{d}{z}} r_1(\alpha) u^\mu \partial_\mu \alpha + T^{\frac{d-2z+2}{z}} r_2(\alpha) u^\mu v^\nu \mathcal{L}_\beta h_{\mu\nu} \right) + \mathcal{O}(\partial^2), \tag{5.41}$$

exhibiting two NHS transport coefficients in agreement with the statement above.

All together for an uncharged Lifshitz fluid we find 7 dissipative, 1 HS and 2 NHS transport coefficients.

## 5.5 Lorentz boost invariance

What we have obtained is the most general set of first order transport coefficients without assuming boost invariance. In this subsection, we show how to recover relativistic first order hydrodynamics from our results. In Landau frame and on Minkowski spacetime the relativistic energy-momentum tensor is described by (see e.g. [2] for a review of relativistic hydrodynamics)

$$T^\mu{}_\nu = \tilde{\mathcal{E}} U^\mu U_\nu + P \Pi^\mu{}_\nu - \zeta \Pi^\mu{}_\nu \partial_\rho U^\rho - \eta \Pi^{\mu\rho} \Pi_\nu{}^\sigma \Sigma_{\rho\sigma} , \tag{5.42}$$

where $\zeta$ and $\eta$ are the bulk and shear viscosity terms, respectively, which are independent of $v^2$ in the Lorentzian case. A relativistic fluid is characterized by a velocity

$$U^\mu = \gamma(1, v^i), \qquad U_\mu = \gamma(-1, v^i) , \tag{5.43}$$

where $\gamma = (1-v^2)^{-1/2}$. In writing the relativistic energy-momentum tensor (5.42), we defined the projector $\Pi^\mu{}_\nu = \delta^\mu_\nu + U^\mu U_\nu$, while the shear tensor $\Sigma_{\rho\sigma}$ is given by

$$\Sigma_{\rho\sigma} = \partial_\rho U_\sigma + \partial_\sigma U_\rho - \frac{2}{d} \eta_{\rho\sigma} \partial_\lambda U^\lambda , \tag{5.44}$$

where $\eta_{\mu\nu} = \mathrm{diag}(-1, 1, \ldots, 1)$ is the $d$-dimensional Minkowski metric.

A necessary condition for this to be recovered from our general framework is the boost Ward identity, $T^0_{(1)i} = -T^i_{(1)0}$, where $T^i_{(1)0} = -v^j T^i_{(1)j}$ in Landau frame. Using (5.8) and (5.9) we see that this translates into the requirements

$$v^j \eta^{ijkl} = \kappa^{kli}, \qquad v^j \kappa^{ijk} = \kappa_{ik} . \tag{5.45}$$

This implies the relations

$$f_3 = v^2 f_2, \qquad f_1 = v^2 f_3, \qquad s_5 = v^2 s_6, \qquad s_4 = v^2 s_2, \qquad s_1 = v^2 s_4 . \tag{5.46}$$

Hence in Landau frame it is sufficient to compare our expression for $T^i_{(1)j}$ with (5.42) at first order. A tedious calculation shows that the two expressions agree if and only if

$$
\begin{aligned}
s_3 &= -\zeta\gamma - \frac{2}{d(d-1)}\eta\gamma , \\
s_6 &= -\zeta\gamma^3 + \frac{2}{d}\eta\gamma^3 , \\
s_2 &= -\zeta\gamma^5 - 2\eta\gamma^5 + \frac{2}{d}\eta\gamma^5 , \\
f_2 &= -\eta\gamma^3 , \\
\mathfrak{t} &= -\eta\gamma ,
\end{aligned}
\tag{5.47}
$$

so that we recover the standard transport coefficients $\zeta$ and $\eta$. In this way, it is also possible to recover (massless) Galilean invariant fluids, although we refrain from giving the details.

## 5.6 Small velocity limit

We will assume that the transport coefficients are functions of $T$ and $v^2$ which admit a Taylor expansion in $v^2$. Demanding that the tensors $\kappa_{jk}$, $\eta_{jkl}$, $\eta^{ijkl}$ and $\kappa^{ijk}$ have a regular limit for $v^2 = 0$ leads to the conditions

$$
\begin{aligned}
f_1 &= \mathcal{O}(v^2), \\
s_1 - f_1 &= \mathcal{O}(v^4),
\end{aligned}
$$

$$\begin{aligned}
f_3 &= \mathcal{O}(v^2), \\
f^{\mathrm{NHS}} &= \mathcal{O}(v^2), \\
s_5 &= \mathcal{O}(v^2), \\
s_1^{\mathrm{NHS}} &= \mathcal{O}(v^2), \\
s_4 &= \mathcal{O}(v^2), \\
s_2^{\mathrm{NHS}} &= \mathcal{O}(v^2), \\
\mathfrak{t} &= \mathcal{O}(1), \\
s_3 - \frac{2}{d-1}\mathfrak{t} &= \mathcal{O}(1), \\
f_2 - \mathfrak{t} &= \mathcal{O}(v^2), \\
\frac{2}{d-1}\mathfrak{t} - s_3 + s_6 &= \mathcal{O}(v^2), \\
s_3^{\mathrm{NHS}} &= \mathcal{O}(v^2), \\
s_2 - s_6 - 2f_2 &= \mathcal{O}(v^2).
\end{aligned} \tag{5.48}$$

In obtaining these expressions, we have used that if $A + B = \mathcal{O}(v^2)$ and $A = \mathcal{O}(v^2)$, then $B = \mathcal{O}(v^2)$.

Substituting these results into the relations (5.11)–(5.14) leads to

$$\kappa_{jk} = \frac{f_1}{v^2}\delta_{jk} + \mathcal{O}(v^2), \quad \eta^{ijkl} = \mathfrak{t}(\delta^{il}\delta^{jk} + \delta^{ik}\delta^{jl}) + \left(s_3 - \frac{2\mathfrak{t}}{d-1}\right)\delta^{ij}\delta^{kl} + \mathcal{O}(v^2), \quad (5.49)$$

and $\eta_{jkl} = \kappa^{ijk} = 0$, where it is particularly noteworthy that all NHS transport drops out. We further remark that the Lorentzian case – which we recovered from our general framework in Section 5.5 – satisfies these conditions, and in particular some of these coefficients exhibit small-$v^2$ behavior involving even higher powers of velocity, e.g. $f_1 = \mathcal{O}(v^4)$.

It is interesting to look at the leading order terms in the expansion of $T^0_{(1)j}$ and $T^i_{(1)j}$ in powers of $v^2$, where $v^2$ is small compared to the speed of sound. These terms are $\mathfrak{t}$, $\frac{s_3}{2} - \frac{\mathfrak{t}}{d-1}$ and $f_1$. These are the shear and bulk viscosity and a new transport coefficient $\varpi$ (denoted by $\pi$ in [11]), i.e.

$$\mathfrak{t} = -\eta + \mathcal{O}(v^2), \qquad \frac{s_3}{2} - \frac{\mathfrak{t}}{d-1} = \frac{1}{d}\eta - \frac{1}{2}\zeta + \mathcal{O}(v^2), \qquad f_1 = -\varpi v^2 + \mathcal{O}(v^4). \tag{5.50}$$

Hence to leading order in $v^i$ we find

$$T^0_{(1)j} = -\varpi\partial_t v^j + \ldots, \tag{5.51}$$

$$T^i_{(1)j} = -\zeta\delta_{ij}\partial_k v^k - \eta(\partial_i v_j + \partial_j v_i - \frac{2}{d}\delta^i_j\partial_k v^k) + \ldots, \tag{5.52}$$

where $\zeta$, $\eta$ and $\varpi$ are non-negative (as follows from the results of Section 5.3). Thus, we recover the results of [11].

In order to make further contact with the results of [11], we turn to the hydrostatic sector – in particular, the transport coefficients $\varpi$ and $\zeta$ that appear in (5.51) and (5.52) can be split into dissipiative and hydrostatic non-dissipative parts[16],

$$\varpi = \varpi_{\mathrm{D}} + \varpi_{\mathrm{HS}}, \qquad \zeta = \zeta_{\mathrm{D}} + \zeta_{\mathrm{HS}}. \tag{5.53}$$

---

[16]Since we showed in (5.49) that all NHS transport drops out in the limit of small background velocity, we ignore that sector in the split.

Using our result (4.42), we see that only the terms involving $\mathcal{J}_1$, $\mathcal{J}_7$ and $\mathcal{J}_3$ survive the small velocity limit. Via the relation (2.21), we find that the terms involving $F_2$ in $\mathcal{J}_7$ and $\mathcal{J}_3$ cancel. Thus, when expanding around $v^i = \delta v^i$ on a flat background, we find that

$$T^0_{(1)\mathrm{HS}i} \;=\; -\frac{\rho_0 F_1(T_0)}{s_0}\partial_t \delta v^i = -\varpi_{\mathrm{HS}}\partial_t \delta v^i \,, \tag{5.54}$$

$$T^i_{(1)\mathrm{HS}j} \;=\; \delta^i_j F_1(T_0)\frac{\partial P_0}{\partial s_0}\partial_k \delta v^k = -\zeta_{\mathrm{HS}}\delta^i_j \partial_k \delta v^k \,, \tag{5.55}$$

where the subscript '0' – e.g. in $T_0$ – denotes the value of the corresponding variable in the global equilibrium around which we expand. In [11], it is shown that

$$\varpi_{\mathrm{HS}} \;=\; -a_T \frac{\rho_0 T_0}{s_0} \,, \tag{5.56}$$

$$\zeta_{\mathrm{HS}} \;=\; a_T T_0 \frac{\partial P_0}{\partial s_0} \,, \tag{5.57}$$

where $a_T$ is the derivative with respect to temperature of a certain hydrostatic transport coefficient $a$ that was identified in [11]. Comparing these with our expressions in (5.54) and (5.55) gives the relation

$$F_1(T_0) \;=\; -a_T T_0 \,, \tag{5.58}$$

thereby identifying $F_1$ with the function $a$ used in [11].

## 6  Discussion and outlook

In this paper we presented the complete first-order energy-momentum tensor for a boost-agnostic fluid in curved spacetime, going beyond the linearized results obtained in [11]. Implementing the constraint of non-negativity of the divergence of the entropy current, we find 10 dissipative, 2 hydrostatic non-dissipative and 4 non-hydrostatic non-dissipative transport coefficients. In the linearized regime the latter four coefficients vanish as a result of implementing the Onsager relations.

Using the curved spacetime formulation we explicitly obtained all non-dissipative transport coefficients, notably both in Landau frame and Lagrangian frame, by using a Lagrangian whose form was derived by starting with the hydrostatic partition function. Furthermore, we checked that our final results reproduce the well-known relativistic first-order transport coefficients when Lorentz boost symmetries are present. We also treated the special case when the hydrodyanmic theory exhibits an additional Lifshitz scale invariance, in which case there are 7 dissipative, 1 hydrostatic non-dissipative and 2 non-hydrostatic non-dissipative transport coefficients. We also studied the small velocity limit, reproducing the results of [11].

With the full geometrical information at our disposal, we can now for example compute Kubo formulae and relate individual transport coefficients to a particular linear response. Consider for example the response of the system in flat space to a purely time-dependent perturbation $\delta h_{\mu\nu}$. The perfect fluid equations of motion (2.27) remain valid to first order if we also impose $\delta P = \delta(e\rho) = 0$, together with

$$\delta v^i \;=\; -\delta h_{0i} - v^j \delta h_{ji} \,, \tag{6.1}$$

$$\delta(e\mathcal{E}) \;=\; \frac{1}{2}T^{00}\delta h_{00} - \frac{1}{2}T^{ij}\delta h_{ij} \,. \tag{6.2}$$

This induces a certain change $\delta T^{\mu\nu}$ which evaluates to

$$\delta T^{ij} = -P\delta h_{ij} - \frac{1}{2}\delta h_{kk}\rho v^i v^j - \rho v^i \delta h_{0j} - \rho v^j \delta h_{0i} - \rho v^i v^k \delta h_{kj} - \rho v^j v^k \delta h_{ki} , \quad (6.3)$$

$$\delta T^{i0} = -\frac{1}{2}\delta h_{kk}\rho v^i - \rho \delta h_{0i} - \rho v^j \delta h_{ij} , \quad (6.4)$$

$$\delta T^{00} = -\frac{1}{2}\delta h_{kk}\rho . \quad (6.5)$$

From these variations one can read off that there are leading order contributions to the two-point function $\langle T^{\mu\nu}T^{\rho\sigma}\rangle$ which are $\omega$-independent. If we substract these leading order contributions, we find from (5.4) that we expect a contribution proportional to $\omega\eta^{\mu\nu\rho\sigma}$ to the two-point function, and hence a Kubo formula of the type $\eta \sim \lim_{\omega\to 0}\frac{1}{\omega}(\langle TT\rangle - \langle TT\rangle_{\text{leading}})$. This is not yet the complete answer though. First of all, the leading order change in $v^i$ is also relevant and using the explicit expression one can show that while at first order $\langle T^{\mu\nu}T^{00}\rangle$ is indeed proportional to $\eta^{\mu\nu 00}$, $\langle T^{\mu\nu}T^{0i}\rangle$ does not contain any contribution of $\eta$, and

$$\langle T^{\mu\nu}T^{ij}\rangle_{(1)} \sim \eta^{\mu\nu ij} - \eta^{\mu\nu 0(i}v^{j)} . \quad (6.6)$$

In addition, we have not yet included the hydrostatic contribution to the stress tensor. These contributions can in principle be extracted from the analysis in Section 4. Alternatively, one can also try to derive Kubo formulae for the HS coefficients directly from the partition function, which in particular guarantees that the stress tensor will be covariantly conserved [61]. We leave a more detailed analysis of all these issues to future work.

Besides the application to Kubo formulae, the geometrical formulation based on Aristotelian (or absolute) spacetime also paves the way for computing hydrodynamic modes – our framework provides the ideal starting point for such an analysis. One is now provided with the tools to answer questions regarding the stability of the hydrodynamical spectrum at first order in curved spacetime, as was studied for boost invariant systems in [19,62]. Another extension of this work is to consider the inclusion of a $U(1)$ charge current as was initiated in [11,12]. The case of Carroll hydrodynamics requires special treatment and will be the topic of future work [20].

Another worthwhile open direction is to consider the relation to holography for the case of Lifsthiz fluids, for which we have identified the reduced set of first-order transport coefficients. Such fluids were discussed in a holographic context in Refs. [23, 41–44]. Following these works, it would be interesting to study hydrodynamic modes of Lifshitz fluids using quasinormal modes in order to find the extra dissipative and non-dissipative transport coefficients, as was done in [45]. It would furthermore be interesting to see if a damping/overdamping transition as reported in [63,64] could be reproduced. More generally, addressing the question of universal properties obeyed by transport coefficients in holographic setups and the development of a full-fledged fluid/gravity correspondence would be relevant to pursue in this case.

In another direction, it would be worthwhile to examine submanifolds and fluids living on them in the spirit of [24]. In particular, it was shown in the context of Newton–Cartan geometry that the normal projection of $v^\mu$ can be interpreted as the transverse velocity of the submanifold. In the absence of boost symmetry, the role of such a velocity might be more prominent.

Finally, as discussed in the introduction, one often finds broken translation symmetry when considering systems where boosts are absent. Assuming there is a hierarchy of (the absence of) these symmetries, one can apply the results presented in this work. It would be important and interesting to apply the formalism developed in this paper to describe particular physical phenomena in concrete systems that do not exhibit boost symmetries.

## Acknowledgements

We especially thank Stefan Vandoren for discussions and collaboration on non-boost invariant hydrodynamics. We also thank Nick Poovuttikul and Lárus Thorlacius for useful discussions. JdB is supported by the European Research Council under the European Unions Seventh Framework Programme (FP7/2007-2013), ERC Grant agreement ADG 834878. JH is supported by the Royal Society University Research Fellowship "Non-Lorentzian Geometry in Holography" (grant number UF160197). EH is supported by the Royal Society Research Grant for Research Fellows 2017 "A Universal Theory for Fluid Dynamics" (grant number RGF\R1\180017) and gratefully acknowledges the hospitality of Nordita while part of this work was undertaken. NO is supported in part by the project "Towards a deeper understanding of black holes with non-relativistic holography" of the Independent Research Fund Denmark (grant number DFF-6108-00340) and by the Villum Foundation Experiment project 00023086. WS is supported by the Icelandic Research Fund (IRF) via a Personal Postdoctoral Fellowship Grant (185371-051).

## A  Hydrodynamic frame transformations & Landau frame

This appendix briefly discusses the general features of hydrodynamic frame transformations. Consider the energy-momentum tensor to first order, which in a generic frame takes the form,

$$T^{\mu}{}_{\nu} = -\left(\tilde{\mathcal{E}} + P + \rho u^2\right)u^{\mu}\tau_{\nu} + \rho u^{\mu}u^{\rho}h_{\rho\nu} + P\delta^{\mu}_{\nu} - T^{\mu}_{(1)}\tau_{\nu} + T^{\mu\rho}_{(1)}h_{\rho\nu}. \tag{A.1}$$

Redefining $T$ and $u^{\mu}$ as

$$T = T' + \delta T, \qquad u^{\mu} = u'^{\mu} + \delta u^{\mu}, \tag{A.2}$$

with $\tau_{\mu}\delta u^{\mu} = 0$ (in order to preserve the normalization $\tau_{\mu}u^{\mu} = 1$), leads to an energy-momentum tensor of the form

$$T^{\mu}{}_{\nu} = -\left(\tilde{\mathcal{E}}' + P' + \rho'u'^2\right)u'^{\mu}\tau_{\nu} + \rho'u'^{\mu}u'^{\rho}h_{\rho\nu} + P'\delta^{\mu}_{\nu} - \tilde{T}^{\mu}_{(1)}\tau_{\nu} + \tilde{T}^{\mu\rho}_{(1)}h_{\rho\nu}, \tag{A.3}$$

where $\tilde{\mathcal{E}}' = \tilde{\mathcal{E}}(T', u'^2)$ etc., and where

$$\tilde{T}^{\mu}_{(1)} = T^{\mu}_{(1)} + \left(\delta\tilde{\mathcal{E}} + \delta P + \rho'\delta u^2 + u'^2\delta\rho\right)u'^{\mu} + \left(\tilde{\mathcal{E}}' + P' + \rho'u'^2\right)\delta u^{\mu} + \delta P v^{\mu}, \tag{A.4}$$

$$\tilde{T}^{\mu\rho}_{(1)} = T^{\mu\rho}_{(1)} + \rho'u'^{\mu}\delta u^{\rho} + \rho'u'^{\rho}\delta u^{\mu} + u'^{\mu}u'^{\rho}\delta\rho + \delta P h^{\mu\rho}. \tag{A.5}$$

The defining condition for the transformed energy-momentum tensor (A.3) to be in Landau frame is

$$u'^{\nu}T^{\mu}{}_{\nu} = -\tilde{\mathcal{E}}'u'^{\mu} \quad \Leftrightarrow \quad \tilde{T}^{\mu}_{(1)} = \tilde{T}^{\mu\rho}_{(1)}h_{\rho\nu}u'^{\nu} \quad \Leftrightarrow \quad \tilde{T}^{\mu}_{(1)\nu}u'^{\nu} = 0, \tag{A.6}$$

where $\tilde{T}^{\mu}_{(1)\nu} = -\tilde{T}^{\mu}_{(1)}\tau_{\nu} + \tilde{T}^{\mu\rho}_{(1)}h_{\rho\nu}$ which will be the case provided we have

$$T^{\mu}_{(1)\nu}u'^{\nu} = \left(\delta\tilde{\mathcal{E}} + \frac{1}{2}\rho'\delta u^2\right)u'^{\mu} + \left(\tilde{\mathcal{E}}' + P'\right)\delta u^{\mu}. \tag{A.7}$$

This can be solved for $\delta s$ and $\delta u^{\mu}$ (by contracting with $\tau_{\mu}$ and $h_{\mu\rho}u'^{\rho}$) to give

$$u'^{\mu}\delta s + s'\delta u^{\mu} = T^{\mu}_{(1)\nu}\frac{u'^{\nu}}{T'}, \tag{A.8}$$

$$s'\delta u^{\mu} = T^{\sigma}_{(1)\nu}\Pi'^{\mu}{}_{\sigma}\frac{u'^{\nu}}{T'}, \tag{A.9}$$

where $\Pi'^{\mu}{}_{\sigma} = \delta^{\mu}_{\sigma} - u'^{\mu}\tau_{\sigma}$, and where we used $\delta\tilde{\mathcal{E}} + \frac{1}{2}\rho'\delta u^2 = T'\delta s$ and $\tilde{\mathcal{E}}' + P' = T's'$.

In Landau frame, it follows from (A.6) that the divergence of the entropy current (3.5) reads

$$e^{-1}\partial_{\mu}(eS^{\mu}) = \frac{1}{T}T^{\mu\nu}_{(1)}h_{\nu\rho}u^{\rho}\mathcal{L}_u\tau_{\mu} - \frac{1}{2T}T^{\mu\nu}_{(1)}\mathcal{L}_u h_{\mu\nu} + e^{-1}\partial_{\mu}\left(eS^{\mu}_{(1)\text{non}}\right), \tag{A.10}$$

which we can equivalently express as

$$e^{-1}\partial_{\mu}(eS^{\mu}) = -\frac{1}{2T}T^{\mu\nu}_{(1)}\left(\mathcal{L}_u h_{\mu\nu} - h_{\rho\,\nu}u^{\rho}\mathcal{L}_u\tau_{\mu} - h_{\mu\rho}u^{\rho}\mathcal{L}_u\tau_{\nu}\right) + e^{-1}\partial_{\mu}\left(eS^{\mu}_{(1)\text{non}}\right). \tag{A.11}$$

This was used to rewrite the divergence of the entropy current in Section 5.

# B  Non-canonical entropy current from constitutive relations

In this appendix, we derive the form of the non-canonical entropy current from its constitutive relations on flat space.

In Landau frame the divergence of the non-canonical entropy current must obey

$$e^{-1}\partial_{\mu}\left(eS^{\mu}_{\text{non}}\right) = \frac{1}{2T}T^{\mu\nu}_{\text{HS}}\left(\mathcal{L}_u h_{\mu\nu} - h_{\rho\,\nu}u^{\rho}\mathcal{L}_u\tau_{\mu} - h_{\mu\rho}u^{\rho}\mathcal{L}_u\tau_{\nu}\right). \tag{B.1}$$

On flat space and in Cartesian coordinates, $\tau_{\mu}$, $h_{\mu\nu}$ and their inverses $v^{\mu}$ and $h^{\mu\nu}$ are constant and furthermore we can write

$$\partial_{\mu}S^{\mu}_{\text{non}} = \frac{1}{T}T^{\mu}_{\text{HS}\,\nu}\partial_{\mu}u^{\nu} = \frac{1}{2T}T^{\mu\nu}_{\text{HS}}\left(\partial_{\mu}u_{\nu} + \partial_{\nu}u_{\mu}\right), \tag{B.2}$$

where we defined $u_{\nu} = h_{\nu\rho}u^{\rho}$, which is a purely spatial object. The hydrostatic part of the energy-momentum tensor must obey all the usual properties the full energy-momentum tensor obeys. In particular, it must be symmetric in its spatial indices. To avoid clutter, we will drop the subscript HS in this appendix.

The constitutive relations for the non-canonical entropy current and the hydrostatic momentum-stress tensor $T^{\mu\nu}$ take the form

$$S^{\mu}_{(1)\text{non}} = \chi^{\mu\nu\rho}\partial_{\nu}u_{\rho}, \tag{B.3}$$

$$T^{\mu\nu}_{(1)} = \frac{1}{2}\eta^{\mu\nu\rho\sigma}\left(\partial_{\rho}u_{\sigma} + \partial_{\sigma}u_{\rho}\right) + \frac{1}{2}\eta^{\mu\nu\rho\sigma}_{\text{rot}}\left(\partial_{\rho}u_{\sigma} - \partial_{\sigma}u_{\rho}\right), \tag{B.4}$$

at first order in derivatives (compare with (4.42) and note that the additional terms due to torsion and extrinsic curvature are absent on flat space). The decomposition of the tensors can be performed with the help of the tensors $v^{\mu}$, $h^{\mu\kappa}u_{\kappa}$ and $h^{\mu\nu}$. We find

$$\begin{aligned}\chi^{\mu\nu\rho} = {} & e_1 v^{\mu}h^{\nu\rho} + e_2 v^{\mu}h^{\nu\kappa}h^{\rho\lambda}u_{\kappa}u_{\lambda} + e_3 v^{\mu}v^{\nu}h^{\rho\sigma}u_{\sigma} + e_4 h^{\mu\nu}h^{\rho\sigma}u_{\sigma} + e_5 h^{\mu\rho}h^{\nu\sigma}u_{\sigma} \\ & + e_6 h^{\mu\rho}v^{\nu} + e_7 h^{\mu\sigma}h^{\nu\rho}u_{\sigma} + e_8 h^{\mu\kappa}h^{\nu\lambda}h^{\rho\sigma}u_{\kappa}u_{\lambda}u_{\sigma} + e_9 h^{\mu\kappa}v^{\nu}h^{\rho\sigma}u_{\kappa}u_{\sigma}, \end{aligned} \tag{B.5}$$

$$\begin{aligned}\eta^{\mu\nu\rho\sigma}_{\text{rot}} = {} & c_1\left(v^{\mu}h^{\nu\rho}h^{\sigma\kappa}u_{\kappa} + v^{\nu}h^{\mu\rho}h^{\sigma\kappa}u_{\kappa} - v^{\mu}h^{\nu\sigma}h^{\rho\kappa}u_{\kappa} - v^{\nu}h^{\mu\sigma}h^{\rho\kappa}u_{\kappa}\right) \\ & + c_2\left(h^{\mu\kappa}h^{\nu\rho}h^{\sigma\lambda}u_{\kappa}u_{\lambda} + h^{\nu\kappa}h^{\mu\rho}h^{\sigma\lambda}u_{\kappa}u_{\lambda} - h^{\mu\kappa}h^{\nu\sigma}h^{\rho\lambda}u_{\kappa}u_{\lambda} - h^{\nu\kappa}h^{\mu\sigma}h^{\rho\lambda}u_{\kappa}u_{\lambda}\right). \end{aligned} \tag{B.6}$$

In deriving the constitutive relation for $\eta^{\mu\nu\rho\sigma}_{\text{rot}}$, we used the fact that[17] $v^{\rho}\left(\partial_{\rho}u_{\sigma} - \partial_{\sigma}u_{\rho}\right) = v^{\rho}\left(\partial_{\rho}u_{\sigma} + \partial_{\sigma}u_{\rho}\right)$ so that we can choose the last two indices of $\eta^{\mu\nu\rho\sigma}_{\text{rot}}$

---

[17]This is a consequence of being on flat space, since $v^{\mu}\partial_{\nu}u_{\mu} = u_{\mu}\partial_{\nu}v^{\mu} = 0$.

to be spatial. We will leave $\eta^{\mu\nu\rho\sigma}$ unspecified, but it admits the same decomposition as in Section 5.

Equation (B.2) can be written as

$$
\begin{aligned}
\chi^{\mu\nu\rho}\partial_\mu\partial_\nu u_\rho \;=\;& \frac{1}{4T}\eta^{\mu\nu\rho\sigma}\big(\partial_\mu u_\nu+\partial_\nu u_\mu\big)\big(\partial_\rho u_\sigma+\partial_\sigma u_\rho\big) \\
&+\frac{1}{4T}\eta_{\text{rot}}^{\mu\nu\rho\sigma}\big(\partial_\mu u_\nu+\partial_\nu u_\mu\big)\big(\partial_\rho u_\sigma-\partial_\sigma u_\rho\big)-\big(\partial_\mu\chi^{\mu\nu\rho}\big)\partial_\nu u_\rho. \quad \text{(B.7)}
\end{aligned}
$$

We would like to solve this equation on shell. By 'solving' we mean finding expressions for the coefficients $e_1$ to $e_9$. The left hand side only contains second order derivative terms, while the right hand side only contains products of first order derivative terms. However, on shell these are not independent, as we now discuss.

On flat space the equation of motion of the perfect fluid part, see equations (2.36), can be written as

$$
\partial_\rho \log T = -\frac{1}{2}X_\rho{}^{\mu\nu}\big(\partial_\mu u_\nu+\partial_\nu u_\mu\big), \quad \text{(B.8)}
$$

where $X_\rho{}^{\mu\nu}$ is defined in (2.37). If we differentiate both sides with respect to $\partial_\sigma$ and anti-symmetrize in $\rho$ and $\sigma$, we obtain a relation among second order derivatives and products of first order derivatives. We seek a scalar equation like the left hand side of (B.7), and the only way to obtain such an expression is to contract the curl of (B.8) with $v^\rho u^\sigma$. This leads to

$$
Y^{\mu\nu\rho}\partial_\mu\partial_\nu u_\rho = \frac{1}{2}v^\rho u^\sigma\big(\partial_\rho X_\sigma{}^{\mu\nu}-\partial_\sigma X_\rho{}^{\mu\nu}\big)\big(\partial_\mu u_\nu+\partial_\nu u_\mu\big), \quad \text{(B.9)}
$$

where $Y^{\mu\nu\rho}$ is defined by

$$
Y^{\mu\nu\rho} = (v^\sigma u^\mu - v^\mu u^\sigma)X_\sigma{}^{\nu\rho}. \quad \text{(B.10)}
$$

The condition that (B.7) does not contain any second order derivatives can be fulfilled in two different ways. The first is to take $\chi^{\mu\nu\rho}$ proportional to $Y^{\mu\nu\rho}$. The second option is to take a $\chi^{\mu\nu\rho}$ that is anti-symmetric in its first two indices $\chi^{\mu\nu\rho}=-\chi^{\nu\mu\rho}$. Using (B.5) the latter is of the form

$$
\chi^{\mu\nu\rho} = 2e_1 v^{[\mu}h^{\nu]\rho}+2e_2 v^{[\mu}h^{\nu]\kappa}h^{\rho\lambda}u_\kappa u_\lambda+2e_5 h^{\rho[\mu}h^{\nu]\kappa}u_\kappa, \quad \text{(B.11)}
$$

where we have set the symmetric part $\chi^{(\mu\nu)\rho}=0$. This latter condition leads to $e_6=-e_1$, $e_3=e_4=e_8=0$, $e_7=-e_5$ and $e_9=-e_2$. We thus conclude that at this stage the most general form of $\chi^{\mu\nu\rho}$ is

$$
\chi^{\mu\nu\rho} = f\,Y^{\mu\nu\rho}+2e_1 v^{[\mu}h^{\nu]\rho}+2e_2 v^{[\mu}h^{\nu]\kappa}h^{\rho\lambda}u_\kappa u_\lambda+2e_5 h^{\rho[\mu}h^{\nu]\kappa}u_\kappa, \quad \text{(B.12)}
$$

where $f$ is an arbitrary function. Compared to (B.5) we are now using a slightly different parameterization, namely the right hand side of (B.5) has been applied to $\chi^{\mu\nu\rho}-f\,Y^{\mu\nu\rho}$ after which we impose that the result is antisymmetric in $\mu$ and $\nu$. We hope that this will not cause any confusion.

Equation (B.7) has now been reduced to an equation involving only products of first order derivatives and can be written as

$$
\begin{aligned}
0 \;=\;& -\frac{f}{2}v^\rho u^\sigma\big(\partial_\rho X_\sigma{}^{\mu\nu}-\partial_\sigma X_\rho{}^{\mu\nu}\big)\big(\partial_\mu u_\nu+\partial_\nu u_\mu\big)+\frac{1}{4T}\eta^{\mu\nu\rho\sigma}\big(\partial_\mu u_\nu+\partial_\nu u_\mu\big)\big(\partial_\rho u_\sigma+\partial_\sigma u_\rho\big) \\
&+\frac{1}{4T}\eta_{\text{rot}}^{\mu\nu\rho\sigma}\big(\partial_\mu u_\nu+\partial_\nu u_\mu\big)\big(\partial_\rho u_\sigma-\partial_\sigma u_\rho\big)-\big(\partial_\mu\chi^{\mu\nu\rho}\big)\partial_\nu u_\rho. \quad \text{(B.13)}
\end{aligned}
$$

The first three terms do not contain any term of the form $X^{\mu\nu\rho\sigma}\big(\partial_\mu u_\nu-\partial_\nu u_\mu\big)\big(\partial_\rho u_\sigma-\partial_\sigma u_\rho\big)$, where $X^{\mu\nu\rho\sigma}$ is purely spatial, i.e. all contractions with $\tau_\mu$ give zero. Any such term arising

from $\left(\partial_\mu \chi^{\mu\nu\rho}\right)\partial_\nu u_\rho$ must therefore have a vanishing coefficient. Such a term does indeed arise: its coefficient is $e_5$, and so we conclude that $e_5 = 0$.

Using that $e_5 = 0$ and the form of $\chi$ derived in (B.12), we conclude that at this stage the most general allowed non-canonical entropy current (B.3) is

$$S^\mu_{(1)\text{non}} = \left(f\, Y^{\mu\nu\rho} + 2e_1 v^{[\mu} h^{\nu]\rho} + 2e_2 v^{[\mu} h^{\nu]\kappa} h^{\rho\lambda} u_\kappa u_\lambda\right)\partial_\nu u_\rho\,. \tag{B.14}$$

Using equations (B.10) and (B.8), this can also be written as

$$S^\mu_{(1)\text{non}} = \frac{f}{T}\left(v^\mu u^\nu - v^\nu u^\mu\right)\partial_\nu T + e_1 \partial_\nu\left(v^\mu u^\nu - v^\nu u^\mu\right) + \frac{e_2}{2}\left(v^\mu u^\nu - v^\nu u^\mu\right)\partial_\nu v^2\,, \tag{B.15}$$

where we also used that $u^\mu = -v^\mu + h^{\mu\kappa}u_\kappa$ and that the background tensors are assumed to be constant. Moreover, we remind the reader that $u^\mu = (1, v^i)$, so that we may write $u^2 = v^2$. We now use the freedom that we can add an identically conserved current to the entropy current, as we are not classifying such terms. We can thus write

$$
\begin{aligned}
S^\mu_{(1)\text{non}} &= \frac{f}{T}\left(v^\mu u^\nu - v^\nu u^\mu\right)\partial_\nu T + e_1 \partial_\nu\left(v^\mu u^\nu - v^\nu u^\mu\right) + \frac{e_2}{2}\left(v^\mu u^\nu - v^\nu u^\mu\right)\partial_\nu v^2 \\
&\quad + \partial_\nu\left[g\left(v^\mu u^\nu - v^\nu u^\mu\right)\right]\,,
\end{aligned} \tag{B.16}
$$

for any function $g$. If we then choose $g$ such that $\frac{\partial g}{\partial T} = -\frac{f}{T}$ we obtain the non-canonical entropy current

$$S^\mu_{(1)\text{non}} = G \partial_\nu\left(v^\mu u^\nu - v^\nu u^\mu\right) - \frac{F}{v^2}\left(v^\mu u^\nu - v^\nu u^\mu\right)\partial_\nu v^2\,, \tag{B.17}$$

where $G = e_1 + g$ and $F = -\frac{v^2 e_2}{2} - v^2 \frac{\partial g}{\partial v^2}$. We thus recover the result (4.72) written in flat space.

The divergence of (B.17) must obey

$$\partial_\mu S^\mu_{(1)\text{non}} = \frac{1}{4T}\eta^{\mu\nu\rho\sigma}\left(\partial_\mu u_\nu + \partial_\nu u_\mu\right)\left(\partial_\rho u_\sigma + \partial_\sigma u_\rho\right) + \frac{1}{4T}\eta^{\mu\nu\rho\sigma}_{\text{rot}}\left(\partial_\mu u_\nu + \partial_\nu u_\mu\right)\left(\partial_\rho u_\sigma - \partial_\sigma u_\rho\right)\,. \tag{B.18}$$

We can solve this equation for the coefficients $c_1$ and $c_2$ in (B.6) as well as for those that appear in the symmetric part of $\eta^{\mu\nu\rho\sigma} = \eta^{\rho\sigma\mu\nu}$ in terms of $F$ and $G$, but we will refrain from giving the explicit result. Instead we refer to Section 4 for such expressions.

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
