# Peer review of "Non-Boost Invariant Fluid Dynamics"

_SciPost Physics, doi:SciPost Phys. 9, 018 (2020)_

## Round 1 · Referee Report · Anonymous (Referee 1) · 2020-6-15

Report

The authors do an exhaustive classification of first order transport coefficients in fluids which may lack boost invariance and study how they are constrained in some particular cases, imposing Onsager relations, with Lifshitz scale invariance or with Lorentz boost invariance. The results could be of interest in systems with broken boost invariance that appear in condensed matter or in general for fluids in contact with an external medium that sets a fixed reference frame.

The paper completes the classification of possible transport coefficients of fluids of this type to first order (assuming there is no parity breaking), introducing a new set of coefficients that are non-linear in the velocities and would be absent in more symmetric fluids. The analysis is carefully done and the results could be interesting for a variety of researchers working on effective field theories and hydrodynamics.

I have just two small comments/questions that do not affect to the main content of the paper:

-In the introduction the authors state that a superfluid "that appears in nature" would have an expectation value for a gauge potential. I'm not sure what do they have in mind exactly. To mention some cases that I'm sure are known to the authors, one could have a superfluid at zero density if one has a complex scalar with a Higgs-like potential. Similarly, one could have a superfluid at finite density just by introducing a chemical potential for a complex scalar, without gauge fields ever entering the picture. Regarding real superfluids, Helium can be described as having (spontaneously broken) Galilean boost invariance. So I find this discussion a bit confusing.

-Equation (2.27) is declared as the conservation equation extrapolating from the diffeomorphism Ward identity in the action. Maybe assuming some conditions over $\tau_\mu$, would it be possible to define a conserved charge as one does for instance in other cases as the integral of a density, and derive the local conservation equation of the charge density from this definition?

---

## Round 1 · Referee Report · Anonymous (Referee 2) · 2020-6-23

Report

This paper studies hydrodynamics to first-order in gradients in the case where there is no boost invariance, i.e. theories with a privileged reference frame. In such theories the transport coefficients are functions also of the velocity with respect to that frame.

The novelty of the construction in this paper is to take the uncharged fluid case and generalise it to a curved background. The geometric data is discussed in terms of Aristotelian geometry. Some of the transport coefficients are non-dissipative, and the authors pay close attention to this fact and go as far as to construct an action for these theories.

I recommend that the paper be published as it is an important contribution to the area, particularly for their detailed analysis of how to couple to background geometry in this context, as well as their treatment of the hydrostatic partition function. However I have some minor points that should be addressed first:

  1. In the relativistic fluid context, coupling to background geometry does not introduce new transport coefficients at first order. One covariantizes the first order expressions, but new transport coefficients appear at second order in gradients with a curvature term. It would be useful for the authors to clarify whether this is true also in the non-boost-invariant setting. Does the curved background allow for new transport coefficients at first hydrodynamic order, as compared with flat space? Indeed, the authors do state that curvatures first appear at second order in gradients, however I think some further discussion of this point in the manuscript would help clarify the significance of the calculations they have presented.

  2. If the answer to the question in point 1. above is negative, then all the transport coefficients studied by the authors should be enumerated previously in [12]. The authors do acknowledge agreement with [12] at the level of the constitutive relations, however I think it would be valuable for readers if they could provide the explicit relations between the transport coefficients appearing in their constitutive relations and the existing definitions.

---

## Round 1 · Referee Report · Anonymous (Referee 3) · 2020-7-2

Report

The paper presents a complete and detailed analysis of the hydrodynamics of a fluid without any boost symmetry. They use the method of coupling the system to an external background geometry, analogously to the Newton-Cartan geometry that has been used to derive Lifshitz hydrodynamics. They derive the complete structure of transport coefficients at first order in derivatives but non-linearly in the fluctuations. In addition, the constraints given by the positivity of entropy production are analyzed. The exposition is clear and all the steps of the derivation are given. The results are interesting and potentially useful, even though at present it is not clear if the formalism has any specific application.

I only have a few minor points to raise: 1) I could not completely follow the logic of the derivation of the conservation equation, (4.25), from the action. Before (4.23), it is stated that the fluid equations of motion follow from a diffeo transformation of the action, but then later it is stated that they follow from varying only \beta^\mu. Perhaps the author can elaborate more on this point.

2) There is an ambiguity in separating the HS from the NHS coefficients, as the NHS terms can be seen as solutions of the homogeneous part of eq. (3.11). However in section (4.4) they are identified with parts of the tensors having specific symmetry properties, symmetric or antisymmetric, so seemingly a specific choice has been made but it’s not clear to me how the ambiguity was fixed.

3) The full energy-momentum tensor can be decomposed in a vector and a tensor part, as in (2.24). Can the transport coefficients be separated according to this decomposition?

---

## Round 4 · Referee Report · Anonymous (Referee 1) · 2020-7-10

Report

The authors have answered satisfactorily the points raised in the report. The results of the paper can potentially have multiple aplications in systems with broken boost invariance, and satisfies the general criteria regarding its presentation. I recommend its publication without need for further changes.

---

## Round 4 · Referee Report · Anonymous (Referee 3) · 2020-7-13

Report

The authors have addressed satisfactorily the concerns raised in the report. I recommend the paper for publication.

---

## Round 4 · Referee Report · Anonymous (Referee 2) · 2020-7-17

Report

In order to avoid the proliferation of conventions and definitions, it would have been useful for the authors to phrase the constraints that they have found in terms of the 16 transport coefficients that were already defined in the literature. These are given (explicitly and in closed form) in equation 2.24 of reference [12] (after taking the neutral limit).

However I do not wish to further belabour this point and delay publication of the paper. I can recommend that the paper now be published.

---

## Round 4 · Author Response

We thank the referees for their time. In this letter we respond to comments and suggestions of the referee reports point by point. We are confident that the improvements will satisfy the referee and will increase the quality of the paper.

  • First point of report 1: We agree that the formulation here is confusing and it was by no means intended as a general statement about superfluids. We therefore changed the formulation to make it clear that we are merely describing one illustrative example.

  • Second point of report 1: From the conservation equation, by contracting (as usual) with a Killing vector of the (absolute space) background, one can obtain conserved currents giving rise to conserved charges. While it is not essential for the rest of the paper, we have added a footnote around eq.(2.27) pointing this out, as an aid to the reader.

  • First point of report 2: As is also the case in relativistic fluids, in the non-boost invariant case we similarly do not get any new transport coefficients at first hydrodynamic order as a consequence of considering the background to be curved (viz also the statements around eq. (2.18)). This should also be clear from the results of Section 5, where we restrict to a flat background. We believe the utility of considering the theory on a curved background should be apparent from the covariance entering the computations and results of the paper. We have added a remark in the introduction stressing the first point above, namely that the number of transport coefficients is unaffected by the introduction of background curvature to first order in derivatives.

  • Second point of report 2: Our method of analysis diverges in fact from [12] already at an earlier stage. We agree that there are 16 derivative structures to begin with. Two of these 16 only appear in the analysis of the HS part of the energy-momentum tensor (they appear in $\eta_{\text{rot}}$). The general form of the constitutive relations concern the decomposition of what we call the eta tensor. Non-negativity of entropy production restricts the symmetry properties of the eta tensor. We only implement constitutive relations after taking these symmetry properties into account which reduces the number of D and NHS coefficients from 16 down to 14. This by itself makes it already challenging to compare the results. Furthermore, given the fact that the paper [12] does not provide explicit final results for the most general non-boost invariant case (these have to be extracted from Mathematica files) we believe it is beyond our responsibility to do such a comparison. Should paper [12] be updated to present the results in an explicit way, such that we can make a meaningful comparison, we are happy to do so.

  • First point of report 3: What we described is the distinction between the consequence of diffeomorphism invariance depending on whether one considers the theory off-shell or on-shell. Off-shell we get the Ward identity given in (4.24) and when one considers the theory on-shell (i.e. obey the fluid equations of motion) this reduces to the one given in (4.25). We believe that we already added this distinction to the accompanying text around these equations.

  • Second point of report 3: There is indeed an ambiguity in the HS terms since, as far as the antisymmmetric $\eta$-tensor is concerned, such terms can both be in the HS and NHS part. We have added a remark after (3.11) making this more clear from the outset. Moreover, we also point out that we will make a specific (convenient) choice in fixing this ambiguity. This choice was already stated at the end of Section 4.4 (last part of paragraph below eq. (4.110)).

-Third point of report 3: The answer to this question is negative. The reason being that as we are in Landau frame the energy current (vector) is fixed in terms of the momentum-stress tensor. Thus there is no such separation of transport coefficients.

---

## Round 4 · List of Changes

• Added the following sentence to the introduction (top of page 3): "We also show that the number of transport coefficients is unaffected by the introduction of background curvature to first order in derivatives"

  • Adapted text in the introduction regarding the connection to superfluids (bottom of page 4)

  • Added footnote 7 on page 8, showing that contracting Eq. (2.27) with a vector $K^\rho$ gives rise to a conserved current if $K^\rho$ is Killing in the Aristotelian sense we define later.

  • Added a remark below Eq. (3.11) noting that HS terms are only defined up to the addition of NHS terms.

---

## Editorial Decision

published